# Learning to Play No-Press Diplomacy
# with Best Response Policy Iteration

**Thomas Anthony**∗, **Tom Eccles**∗, **Andrea Tacchetti, János Kramár, Ian Gemp,**
**Thomas C. Hudson, Nicolas Porcel, Marc Lanctot, Julien Pérolat, Richard Everett,**
**Roman Werpachowski, Satinder Singh, Thore Graepel and Yoram Bachrach**

DeepMind

## Abstract

Recent advances in deep reinforcement learning (RL) have led to considerable
progress in many 2-player zero-sum games, such as Go, Poker and Starcraft. The
purely adversarial nature of such games allows for conceptually simple and prin-
cipled application of RL methods. However real-world settings are many-agent,
and agent interactions are complex mixtures of common-interest and competitive
aspects. We consider Diplomacy, a 7-player board game designed to accentuate
dilemmas resulting from many-agent interactions. It also features a large com-
binatorial action space and simultaneous moves, which are challenging for RL
algorithms. We propose a simple yet effective approximate best response operator,
designed to handle large combinatorial action spaces and simultaneous moves. We
also introduce a family of policy iteration methods that approximate fictitious play.
With these methods, we successfully apply RL to Diplomacy: we show that our
agents convincingly outperform the previous state-of-the-art, and game theoretic
equilibrium analysis shows that the new process yields consistent improvements.

## 1 Introduction

Artificial Intelligence methods have achieved exceptionally strong competitive play in board games
such as Go, Chess, Shogi [108, 110, 20, 109], Hex [2], Poker [85, 17] and various video games [63, 84,
60, 94, 45, 120, 55, 14]. Despite the scale, complexity and variety of these domains, a common focus
in multi-agent environments is the class of 2-player (or 2-team) zero-sum games: "1 vs 1" contests.
There are several reasons: they are polynomial-time solvable, and solutions both grant worst-case
guarantees and are interchangeable, so agents can approximately solve them in advance [121, 122].
Further, in this case conceptually simple adaptations of reinforcement learning (RL) algorithms often
have theoretical guarantees. However, most problems of interest are not purely adversarial: e.g. route
planning around congestion, contract negotiations or interacting with clients all involve compromise
and consideration of how preferences of group members coincide and/or conflict. Even when agents
are self-interested, they may gain by coordinating and cooperating, so interacting among diverse
groups of agents requires complex reasoning about others' goals and motivations.

We study **Diplomacy** [19], a 7-player board game. The game was specifically designed to emphasize
tensions between competition and cooperation, so it is particularly well-suited to the study of learning
in mixed-motive settings. The game is played on a map of Europe partitioned into provinces. Each
player controls multiple units, and each turn *all* players move *all* their units simultaneously. One unit
may support another unit (owned by the same or another player), allowing it to overcome resistance
by other units. Due to the inter-dependencies between units, players must coordinate the moves of
their own units, and stand to gain by coordinating their moves with those of other players. Figure 1
depicts interactions among several players (moving and supporting units to/from provinces); we
explain the basic rules in Section 2.1. The original game allows cheap-talk negotiation between

players before every turn. In this paper we focus on learning strategic interactions in a many-agent setting, so we consider the popular *No Press* variant, where no explicit communication is allowed.

Diplomacy is particularly challenging for RL agents. First, it is a *many-player* ($n > 2$) game, so methods cannot rely on the simplifying properties of 2-player zero-sum games. Second, it features *simultaneous moves*, with a player choosing an action without knowledge of the actions chosen by others, which highlights reasoning about opponent strategies. Finally, Diplomacy has a *large combinatorial action space*, with an estimated game-tree size of $10^{900}$, and $10^{21}$ to $10^{64}$ legal joint actions *per turn*. [1]

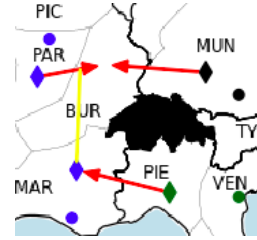

Consequently, although Diplomacy AI has been studied since the 1980s [65, 67], until recently progress has relied on handcrafted rule-based systems, rather than learning. Paquette et al. [90] achieved a major breakthrough: they collected a dataset of $\sim 150,000$ human Diplomacy games, and trained an agent, *DipNet*, using a graph neural network (GNN) to imitate the moves in this dataset. This agent defeated previous state-of-the-art agents conclusively and by a wide margin. This is promising, as imitation learning can often be a useful starting point for RL methods.

Figure 1: Simple example of interactions between several players' moves.

However, to date RL has not been successfully applied to Diplomacy. For example, Paquette et al. [90] used A2C initialised by their imitation learning agent, but this process did not improve performance as measured by the Trueskill rating system [50]. This is unfortunate, as without agents able to optimise their incentives, we cannot study the effects of mixed-motives on many-agent learning dynamics, or how RL agents might account for other agents' incentives (e.g. with Opponent Shaping [36]).

**Our Contribution:** We train RL agents to play No-Press Diplomacy, using a policy iteration (PI) approach. We propose a simple yet scalable improvement operator, *Sampled Best Responses* (SBR), which effectively handles Diplomacy's large combinatorial action space and simultaneous moves. We introduce versions of PI that approximate iterated best response and fictitious play (FP) [16, 97] methods. In Diplomacy, we show that our agents outperform the previous state-of-the-art both against reference populations and head-to-head. A game theoretic equilibrium analysis shows our process yields consistent improvements. We propose a few-shot exploitability metric, which our RL reduces, but agents remain fairly exploitable. We perform a case-study of our methods in a simpler game, Blotto (Appendix A), and prove convergence results on FP in many-player games (Appendix H).

## 2  Background and Related Work

Game-playing has driven AI research since its inception: work on games delivered progress in search, RL and computing equilibria [101, 44, 61, 35, 13, 99, 102, 115, 104, 41, 34] leading to prominent successes in Chess [20], Go [108, 110], Poker [85, 17], multi-agent control domains [10, 79, 6, 112, 124] and video games [63, 84, 60]. Recent work has also used deep RL in many-player games. Some, such as Soccer, Dota and Capture-the-Flag, focus on two teams engaged in a zero-sum game but are cooperative between members of a team [79, 14, 55]. Others, e.g. Hanabi or Overcooked, are fully-cooperative [37, 53, 75, 11, 21]. Most relevantly, some work covers mixed-motive social dilemmas, with both competitive and collaborative elements [81, 74, 76, 24, 36, 105, 54].

There is little work on large, competitive, many-player settings, known to be harder than their two-player counterparts [23, 26]. The exception is a remarkable recent success in many-player no-limit Poker that defeated human experts [18]. However, it uses expert abstractions and end-game solving to reduce the game tree size. Moreover, in Poker collusion is strictly prohibited and players often fold early in the game until only two remain, which reduces the effects of many-player interactions in practice. In contrast, in Diplomacy 2-player situations are rare and alliances are crucial.

**Diplomacy AI Research:** Diplomacy is a long-standing AI challenge. Even in the simpler No-Press variant, AIs are far weaker than human players. Rule-based Diplomacy agents were proposed in the 1980s and 1990s [65, 46, 64, 66]. Frameworks such as DAIDE [98] DipGame [31] and BANDANA [57] promoted development of stronger rule-based agents [56, 117, 33]. One work applied TD-learning with pattern weights [106], but was unable to produce a strong agent. Negotiation

for Computer Diplomacy is part of the Automated Negotiating Agents Competition [5, 27]. We build on DipNet, the recent success in using a graph neural network to imitate human gameplay [90]. DipNet outperformed previous agents, all rule-based systems, by a large margin. However, the authors found that A2C [83] did not significantly improve DipNet. We replicated this result with our improved network architecture (see Appendix E).

**Related Algorithms:** In this work we present PI algorithms that approximate FP. Neural Fictitious Self-Play (NFSP) [49] and Policy Space Response Oracles (PSRO) [73] are two prior algorithms that use RL in an 'inner-loop' to approximate a BR for FP. NFSP makes use of DQN, which requires a small actions space, to apply it to Diplomacy would require decomposing the action space, e.g. into unit actions. PSRO used a version of Actor-Critic, but requires a full RL training run every iteration, and A2C has previously proven ineffective in Diplomacy. In contrast, we use SBR to search for a BR in each stage-game without an 'inner-loop' RL algorithm.

AlphaZero and Expert Iteration [109, 2] previously applied PI with search in classical board games. Their success motivates using PI in Diplomacy, but because their MCTS requires sequential moves and only ∼ 100s of actions per turn to be effective, they cannot be directly applied in Diplomacy.

## 2.1 No-Press Diplomacy: Summary of Game Rules

We provide an intentionally brief overview of the core game mechanics. For a longer introduction, see [90], and the rulebook [19]. The board is a map of Europe partitioned into provinces; 34 provinces are **supply centers** (SCs, dots in PAR, MUN, MAR, and VEN in Figure 1). Each player controls multiple units of a country. Units capture SCs by occupying the province. Owning more SCs allows a player to build more units; the game is won by owning a majority of the SCs. Diplomacy has *simultaneous moves*: each turn every player writes down orders for all their units, without knowing what other players will do; players then reveal their moves, which are executed simultaneously. The next position is fully determined by the moves and game rules, with no chance element (e.g. dice).

Only one unit can occupy a province, and all units have equal strength. A unit may *hold* (guard its province) or *move* to an adjacent province. A unit may also *support* an adjacent unit to hold or move, to overcome opposition by enemy units. Using Figure 1 as a running example, suppose France orders **move** PAR → BUR; if the unit in MUN **holds** then the unit in PAR enters BUR, but if Germany also ordered MUN → BUR, both units 'bounce' and neither enters BUR. If France wanted to insist on entering to BUR, they can order MAR **support** PAR → BUR, which gives France 2 units versus Germany's 1, so France's move order would succeed and Germany's would not. However, MAR's support can be *cut* by Italy moving PIE → MAR, leading to an equal-strength bounce as before.

This example highlights elements that make Diplomacy unique and challenging. Due to simultaneous move resolution, players must anticipate how others will act and reflect these expectations in their own actions. Players must also use a stochastic policy (mixed strategy), as otherwise opponents could exploit their determinism. Finally, cooperation is essential: Germany would not have been able to prevent France from moving to BUR without Italy's help. Diplomacy is specifically designed so that no player can win on their own without help from other players, so players *must* form alliances to achieve their ultimate goal. In the No-Press variant, this causes pairwise interactions that differ substantially from zero-sum, so difficulties associated with mixed-motive games arise in practice.

## 3 Reinforcement Learning Methods

We adopt a policy iteration (PI) based approach, motivated by successes using PI for perfect information, sequential move, 2-player zero-sum board games [2, 109]. We maintain a neural network policy $\hat{\pi}$ and a value function $\hat{V}$. Each iteration we create a dataset of games, with actions chosen by an improvement operator which uses a previous policy and value function to find a policy that defeats the previous policy. We then train our policy and value functions to predict the actions chosen by the improvement operator and the game results. The initial policy $\hat{\pi}^0$ and value function $\hat{V}^0$ imitate the human play dataset, similarly to DipNet [90], providing a stronger starting point for learning.

Section 3.1 describes SBR, our best response approximation method, tailored to handle the simultaneous move and combinatorial action space of Diplomacy. Section 3.2 describes versions of PI that use SBR to approximate iterated best response and fictitious play algorithms. Our neural network training is an improved version of DipNet, described in Section 3.3 and Appendix C.

## 3.1 Sampled Best Response (SBR)

Our PI methods use best response (BR) calculations as an improvement operator. Given a policy $\pi^b$ defined for all players, the BR for player $i$ is the policy $\pi_i^*$ that maximises the expected return for player $i$ against the opponent policies $\pi_{-i}^b$. A best response may not be a good policy to play as it can be arbitrarily poor against policies other than those it responds to. Nonetheless best responses are a useful tool, and we address convergence to equilibrium with the way we use BRs in PI (Section 3.2).

Diplomacy is far too large for exact best response calculation, so we propose a tractable approximation, Sampled Best Response (SBR, Algorithm 1). SBR makes three approximations: (1) we consider making a single-turn improvement to the policy in each state, rather than a full calculation over multiple turns of the game. (2) We only consider taking a small set of actions, sampled from a candidate policy. (3) We use Monte-Carlo estimates over opponent actions for candidate evaluation.

Consider calculating the value of some action $a_i$ for player $i$ against an opponent policy $\pi_{-i}^b$ (hereafter the *base policy*). Let $T(s, \mathbf{a})$ be the transition function of the game and $V^\pi(s)$ be the state-value function for a policy $\pi$. The 1-turn value to player $i$ of action $a_i$ in state $s$ is given by:

$$Q_i^{\pi^b}(a_i|s) = \mathbb{E}_{a_{-i} \sim \pi_{-i}^b} V_i^{\pi^b}(T(s, (a_i, a_{-i})))$$

We use the value network $\hat{V}$ instead of the exact state-value to get an estimated action-value $\hat{Q}_i^{\pi_i^b}(a_i|s)$.

If the action space were small enough, we could exactly calculate $\arg\max_{a_i} \hat{Q}_i^{\pi_i^b}(a_i|s)$, as a 1-turn best response. However, there are far too many actions to consider all of them. Instead, we sample a set of candidate actions $A_i$ from a *candidate policy* $\pi_i^c(s)$, and only consider these candidates for our approximate best response. Now the strength of the SBR policy depends on the candidate policy's strength, as we calculate an improvement compared to $\pi_i^c$ in optimizing the 1-turn value estimate. Note we can use a different policy $\pi^c$ to the policy $\pi^b$ we are responding to.

The number of strategies available to opponents is also too large, so calculating the 1-turn value of any candidate is intractable. We therefore use Monte-Carlo sampling. Values are often affected by the decisions of other players; to reduce variance we use common random numbers when sampling opponent actions: we evaluate all candidates with the same opponent actions (*base profiles*). SBR can be seen as finding a BR to the sampled base profiles, which approximate the opponent policies.

## 3.2 Best Response Policy Iteration

We present a family of PI approaches tailored to using (approximate) BRs, such as SBR, in a many-agent game; we refer to them collectively as Best Response Policy Iteration (BRPI) algorithms (Algorithm 2). SBR depends on the $\pi^b, \pi^c, v$ (base policy, candidate policy and value function); we can use historical network checkpoints (saved previous network parameters) for these. Different choices give different BRPI algorithms. The simplest version is standard PI with BRs, while others BRPI variants approximate fictitious play.

In the most basic BRPI approach, every iteration $t$ we apply SBR to the *latest* policy $\hat{\pi}^{t-1}$ and value $\hat{V}^{t-1}$ to obtain an improved policy $\pi'$ (i.e. SBR($\pi^c = \hat{\pi}^{t-1}, \pi^b = \hat{\pi}^{t-1}, v = \hat{V}^{t-1}$)). We then sample trajectories of self-play with $\pi'$ to create a dataset, to which we fit a new policy $\hat{\pi}^t$ and value $\hat{V}^t$ using the same techniques used to imitate human data (supervised learning with a GNN). We refer to this as Iterated Best Response (IBR). IBR is akin to applying standard single-agent PI methods in self-play, a popular approach for perfect information, 2-player zero-sum games [113, 103, 109, 2].

However, iteration through exact best responses may behave poorly, failing to converge and leading to cycling among strategies (see Appendix A for an example). Further, in a game with simultaneous moves, deterministic play is undesirable, and best responses are typically deterministic. As a potential remedy, we consider PI algorithms based on Fictitious Play (FP) [16, 116, 77, 48]. In FP, at each iteration all players best respond to the empirical distribution over historical opponent strategies. In 2-player zero-sum, the time average of players' strategies converges to a Nash Equilibrium [16, 97]. In Appendix H, we review theory on many-agent FP, and prove that continuous-time FP converges to a coarse correlated equilibrium in many-agent games. This motivates approximating FP with a BRPI algorithm. We now provide two versions of Fictitious Play Policy Iteration (FPPI) that do this.

The first method, FPPI-1, is akin to NFSP [49]. At iteration $t$, we aim to train our policy and value networks $\hat{\pi}^t, \hat{V}^t$ to approximate the time-average of BRs (rather than the latest BR). With such a

network, to calculate the BR at time $t$, we need an approximate best response to the latest policy network (which is the time-average policy), so use SBR($\pi^b = \hat{\pi}^{t-1}$, $v = \hat{V}^{t-1}$). Hence, to train the network to produce the *average* of BRs so far, at the start of each game we uniformly sample an iteration $d \in \{0, 1, \ldots, t-1\}$; if we sample $d = t-1$ we use the latest BR, and if $d < t-1$ we play a game with the historical checkpoints to produce the historical BR policy from iteration $d$. [2]

FPPI-1 has some drawbacks. With multiple opponents, the empirical distribution of opponent strategies does not factorize into the empirical distributions for each player. But a standard policy network only predicts the per-player marginals, rather than the full joint distribution, which weakens the connection to FP. Also, our best response operator's strength is affected by the strength of the candidate policy and the value function. But FPPI-1 continues to imitate old and possibly weaker best responses, even after we have trained stronger policies and value functions.

In our second variant, FPPI-2, we train the policy networks to predict only the latest BR, and explicitly average historical checkpoints to provide the empirical strategy so far. The empirical opponent strategy up to time $t$ is $\mu^t := \frac{1}{t} \sum_{d<t} \pi^d_{-i}$, to draw from this distribution we first sample a historical checkpoint $d < t$, and then sample actions for all players using the same checkpoint. Player $i$'s strategy at time $t$ should be an approximate best response to this strategy, and the next policy network $\pi^t$ imitates that best response. In SBR, this means we use $\pi^b = \mu^t$ as the base policy.

This remedies the drawbacks of the first approach. The correlations in opponent strategies are preserved because we sample from the same checkpoint for all opponents. More importantly, we no longer reconstruct any historical BRs, so can use our best networks for the candidate policy and value function in SBR, independently of which checkpoints are sampled to produce base profiles. For example, using the latest networks for the candidate policy and value function, while uniformly sampling checkpoints for base profiles, could find stronger best responses while still approximating FP. However, FPPI-2's final time-averaged policy is represented by a mixture over multiple checkpoints.

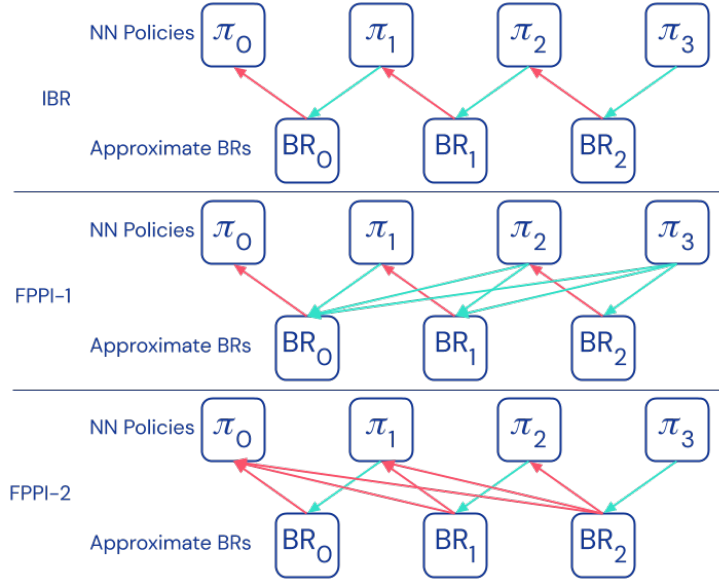

Figure 2: Illustration of Best Response Policy Iteration algorithms. Red lines represent which policies are being best responded against, green lines represent which best responses the neural network imitates

These variants suggest a design space of algorithms combining SBR with PI. (1) The base policy can either be the latest policy (an IBR method), or from a uniformly sampled previous checkpoint (an FP method). (2) We can also use either the latest or a uniformly sampled previous value function. (3) The candidate policy both acts as a regulariser on the next policy and drives exploration, so we consider several options: using the initial (human imitation) policy, using the latest policy, or using a

| **Algorithm 1** Sampled Best Response | **Algorithm 2** Best Response Policy Iteration |
|---|---|
| **Require:** Policies $\pi^b, \pi^c$, value function $v$ | **Require:** Best Response Operator BR |
| 1: **function** SBR($s$:state, $i$:player) | 1: **function** BRPI($\pi_0(\theta), v_0(\theta)$) |
| 2:     **for** $j \leftarrow 1$ to $B$ **do** | 2:     **for** $t \leftarrow 1$ to $N$ **do** |
| 3:         $b_j \sim \pi^b_{-i}(s)$   ▷ Sample Base Profile | 3:         $\pi^{\mathrm{imp}} \leftarrow \mathrm{BR}(\{\pi_j\}_{j=0}^{t-1}, \{v_j\}_{j=0}^{t-1})$ |
| 4:     **for** $j \leftarrow 1$ to $C$ **do** | 4:         $D \leftarrow$ Sample-Trajectories($\pi^{\mathrm{imp}}$) |
| 5:         $c_j \sim \pi^c_i(s)$   ▷ Candidate Action | 5:         $\pi_i(\theta) \leftarrow$ Learn-Policy($D$) |
| 6:         $\hat{Q}(c_j) \leftarrow \frac{1}{B}\sum_{k=1}^{B} v(T(s,(c_j,b_k)))$ | 6:         $v_i(\theta) \leftarrow$ Learn-Value($D$) |
| 7:     **return** $\arg\max_{c \in \{c_j\}_{j=1}^C} \hat{Q}(c)$ | 7:     **return** $\pi_N, v_N$ |

uniformly sampled checkpoint; we also consider mixed strategies: taking half the candidates from initial and half from latest, or taking half from initial and half from a uniformly sampled checkpoint.

Appendix A is a case study analysing how SBR and our BRPI methods perform in a many-agent version of the Colonel Blotto game [15]. Blotto is small enough that exact BRs can be calculated, so we can investigate how exact FP and IBR perform in these games, how using SBR with various parameters affects tabular FP, and how different candidate policy and base policy choices affect a model of BRPI with function approximation. We find that: (1) exact IBR is ineffective in Blotto; (2) stochastic best responses in general, and SBR in particular, improve convergence rates for FP; (3) using SBR dramatically improves the behaviour of IBR methods compared to exact BRs.

### 3.3 Neural Architecture

Our network is based on the imitation learning of DipNet [90], which uses an encoder GNN to embed each province, and a LSTM decoder to output unit moves (see DipNet paper for details). We make several improvements, described briefly here, and fully in Appendix C. (1) We use the board features of DipNet, but replace the original 'alliance features' with the board state at the last moves phase, combined with learned embeddings of all actions taken since that phase. (2) In the encoder, we removed the FiLM layer, and added node features to the existing edge features of the GNN. (3) Our decoder uses a GNN relational order decoder rather than an LSTM. These changes increase prediction accuracy by $4-5\%$ on our validation set (data splits and performance comparison in Appendix C).

## 4 Evaluation Methods

We analyze our agents through multiple lenses: We measure winrates (1) head-to-head between agents from different algorithms and (2) against fixed populations of reference agents. (3) We consider 'meta-games' between checkpoints of one training run to test for consistent improvement. (4) We examine the exploitability of agents from different algorithms. Results of each analysis are in the corresponding part of Section 5.

**Head-to-head comparison:** We play 1v6 games between final agents of different BRPI variants and other baselines to directly compare their performance. This comparison also allows us to spot if interactions between pairs of agents give unusual results. From an evolutionary game theory perspective, 1v6 winrates indicate whether a population of agents can be 'invaded' by a different agent, and hence whether they constitute Evolutionary Stable Strategies (ESS) [114, 111]. ESS have been important in the study of cooperation, as a conditionally cooperative strategies such as Tit-for-Tat can be less prone to invasion than purely co-operative or mostly non-cooperative strategies [4].

**Winrate Against a Population:** We assess how well an agent performs against a reference population. An agent to be evaluated plays against 6 players independently drawn from the reference population, with the country it plays as chosen at random each game. We report the average score of the agent, and refer to this as a "1v6" match. [3] This mirrors how people play the game: each player only ever represents a single agent, and wants to maximize their score against a population of other people. We consider two reference populations: (a) only the DipNet agent [90], the previous state-of-the-art

method, which imitates and hence is a proxy for human play. (b) a uniform mixture of 15 final RL agents, each from a different BRPI method (see Appendix B); BRPI agents are substantially stronger than DipNet, and the mixture promotes opponent diversity.

**Policy Transitivity:** Policy intransitivity relates to an improvement dynamics that cycles through policy space, rather than yielding a consistent improvement in the quality of the agents [52, 9], which can occur because multiple agents all optimize different objectives. We assess policy transitivity with *meta-games* between the checkpoints of a training run. In the meta-game, instead of playing yourself, you elect an 'AI champion' to play on your behalf, and achieve the score of your chosen champion. Each of the seven players may select a champion from among the same set of $N$ pre-trained policies. We randomize the country each player plays, so the meta-game is a symmetric, zero-sum, 7-player game. If training is transitive, choosing later policies will perform better in the meta-game.

Game theory recommends selecting a champion by sampling one of the $N$ champions according to a Nash equilibrium [87], with bounded rationality modelled by a Quantal Response Equilibrium (QRE) [82]. Champions can be ranked according to their probability mass in the equilibrium [9]. [4] We calculate a QRE (see Appendix G) of the meta-game consisting of $i$ early checkpoints, and see how it changes as later checkpoints are added. In transitive runs we expect the distribution of the equilibrium to be biased towards later checkpoints.

Finding a Nash equilibrium of the meta-game is computationally hard (PPAD-complete) [89], so as an alternative, we consider a simplified 2-player meta-game, where the row player's agent plays for one country, and the other player's agent plays in the other 6, we call this the '*1v6 meta-game*'. We report heatmaps of the payoff table, where the row and column strategies are sorted chronologically. If training is transitive, the row payoff increases as row index increases but decreases as the column index increases, which is visually distinctive [8].

**Exploitability:** The exploitability of a policy $\pi$ is the margin by which an adversary (i.e. BR) to $\pi$ would defeat a population of agents playing $\pi$; it has been a key metric for Poker agents [78]. As SBR approximates a BR to its base policy, it can be used to lower bound the base policy's exploitability, measured by the average score of 1 SBR agent against 6 copies of $\pi$. The strongest exploit we found mixes the human imitation policy and $\pi$ for candidates, and uses $\pi$'s value function, i.e. $\text{SBR}(\pi^c = \pi^{\text{SL}} + \pi, \pi^b = \pi, v = V^\pi)$. People can exploit previous Diplomacy AIs after only a few games, so few-shot exploitability may measure progress towards human-level No-Press Diplomacy agents. If we use a 'neutral' policy and value function, and only a few base profiles, SBR acts as a few-shot measure of exploitability. To do this we use the human imitation policy $\pi^{\text{SL}}$ for candidates, and - because $V^{\text{SL}}$ is too weak - a value function from an independent BRPI run $V^{\text{RL}}$.

# 5   Results

We analyse three BRPI algorithms: IBR, FPPI-1 and FPPI-2, defined in Section 3.2. In FPPI-2 we use the latest value function. We sample candidates with a mixture taking half the candidates from the initial policy. The other half for IBR and FPPI-2 is from iteration $t - 1$; for FPPI-1 it comes from the uniformly sampled iteration. At test time, we run all networks at a softmax temperature $t = 0.1$. [5]

**Head-to-head comparison:** we compare our methods to the supervised learning (SL) and RL (A2C) DipNet agents [90]; our SL implementation, with our neural architecture; and Albert [117], the strongest rule-based Diplomacy AI and prior state-of-the-art (see Appendix B for additional comparisons). Table 1 shows the average 1v6 score where a single row agent plays against 6 column agents. For the BRPI methods, these are averaged over 5 seeds of each run. When selecting the column agent, we always use 6 agents from the same training run. This means that, for example, the IBR v IBR match is not symmetric: the row agent is playing against agents it did not train with, whereas the column agents are playing mostly against agents they trained with. This means diagonal entries between BRPI methods should not necessarily equal $\frac{1}{7}$; it tends to give an advantage to column agents.

All of our learning methods give a large improvement over both supervised learning baselines, and an even larger winrate over DipNet A2C. BRPI algorithms are initialised by supervised learning on the same dataset, and begins by calculating a best response, which might have led to a narrow exploit to

the prior SL agents. However, BRPI also shows improved winrates against Albert compared to prior agents, indicating that this is not the case. Among our learning methods, FPPI-2 achieves the best winrate in 1v6 against each algorithm, and is also the strategy against which all singleton agents do the worst.

| | SL [90] | A2C [90] | SL (ours) | FPPI-1 | IBR | FPPI-2 | Albert |
|---|---|---|---|---|---|---|---|
| SL [90] | 14.3% | 7.9% | 16.3% | 3.1% | 1.9% | *1.4%* | 42.0% |
| A2C [90] | 15.5% | 14.3% | 15.5% | 3.0% | 2.3% | *1.3%* | 50.9% |
| SL (ours) | 12.4% | 8.5% | 14.3% | 3.9% | *2.3%* | *2.0%* | 36.3% |
| FPPI-1 | **30.1%** | **27.3%** | **28.9%** | **13.4%** | 7.0% | *6.1%* | **64.5%** |
| IBR | 23.3% | **26.8%** | 24.1% | *12.6%* | 13.6% | *12.6%* | **65.8%** |
| FPPI-2 | 20.4% | 26.4% | 24.7% | *13.5%* | **15.3%** | *13.5%* | **68.7%** |
| Albert | 2.3% | *0.0%* | 3.1% | *0.0%* | *0.0%* | *0.0%* | 14.3% |

Table 1: Average scores for 1 row player vs 6 column players. BRPI methods give an improvement over A2C or supervised learning. All numbers accurate to a $95\%$ confidence interval of $\pm 0.5\%$, except those against Albert which are $\pm 5\%$. Bold numbers are the best value for single agents against a given set of 6 agents, italics are for the best result for a set of 6-agents against each single agent.

**Winrate Against a Population:** The left of Figure 3 shows the performance of our BRPI methods against DipNet through training. Solid lines are the winrate of our agents in 1v6 games (1 BRPI agent vs. 6 DipNet agents), and dashed lines relate to the winrate of DipNet reverse games (1 DipNet agent vs 6 PI agents). The x-axis shows the number of policy iterations. A dashed black line indicates a winrate of 1/7th (expected for identical agents as there are 7 players). The figure shows that all PI methods quickly beat the DipNet baseline before plateauing. Meanwhile, the DipNet winrate drops close to zero in the reverse games. The figure on the right is identical, except the baseline is a uniform mixture of our final agents from BRPI methods. Against this population, our algorithms do not plateau, instead improving steadily through training. The figure shows that FPPI-1 tends to under-perform against our other BRPI methods (FPPI-2 and IBR). We averaged 5 different runs with different random seeds, and display $90\%$ confidence intervals with shaded area.

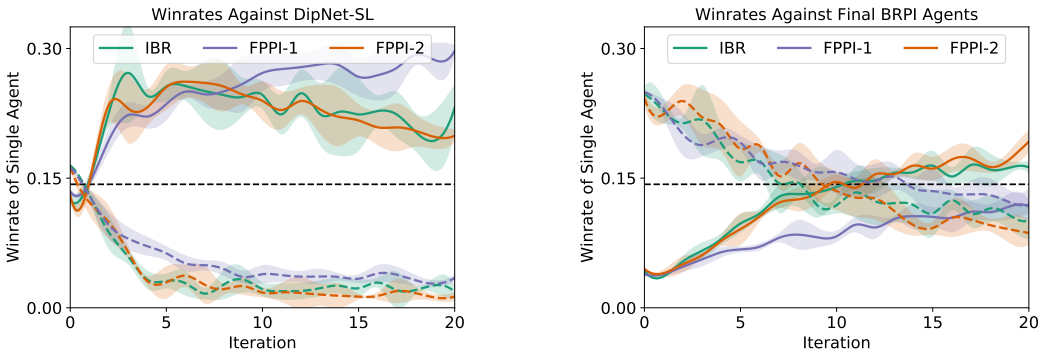

Figure 3: Winrates through training, 1v6 or 6v1 against different reference populations

**Policy Transitivity:** Figure 4 depicts the Diplomacy meta-game between checkpoints produced in a single BRPI run. The heatmaps on the left examine a 1v6 meta game, showing the winrate of one row checkpoint playing against 6 column checkpoints for even numbered checkpoints through the run. The plots show that nearly every checkpoint beats all the previous checkpoints in terms of 1v6 winrate. A checkpoint may beat its predecessors in way that's exploitable by other strategies: checkpoint 4 in the FPPI-2 run beats the previous checkpoints, but is beaten by all subsequent checkpoints by a larger margin than previous checkpoints.

The right side of Figure 4 shows the Nash-league of the full meta-game. The $i^{th}$ row shows the distribution of a QRE in the meta-game over the first $i$ checkpoints analyzed (every row adds the next checkpoint). We consider checkpoints spaced exponentially through training, as gains to further training diminish with time. The figure shows that the QRE consistently places most of the mass on the recent checkpoint, indicating a consistent improvement in the quality of strategies, rather than cycling between different specialized strategies. This is particularly notable for IBR: every

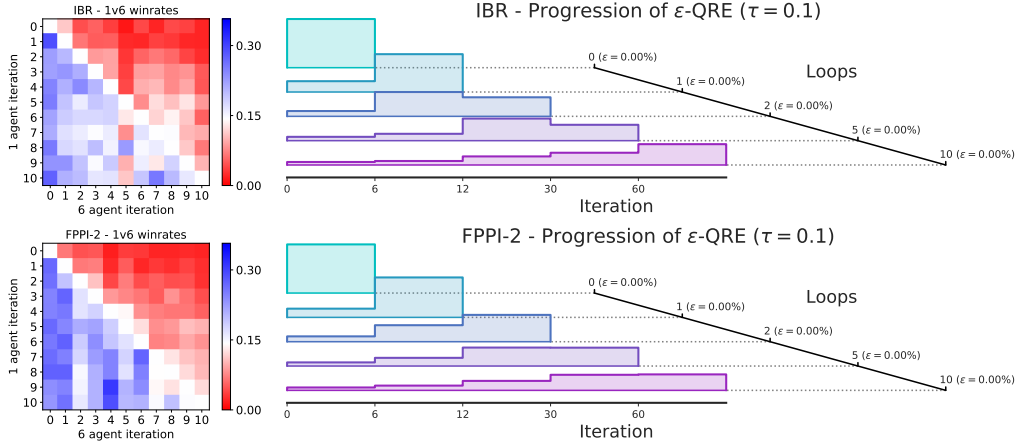

Figure 4: Transitivity Meta-Games: Top: IBR, Bottom: FPPI-2. Left: 1v6, Right: QRE in 7-player.

checkpoint is only trained to beat the previous one, yet we still observe transitive improvements during the run (consistent with our positive findings for IBR in Blotto in Appendix A.5.3).

**Exploitability:** We find that all our agents are fairly exploitable at the end of training (e.g. the strongest exploiters achieve a 48% winrate), and the strongest exploit found does not change much through training, but few-shot exploitability *is* reduced through training. Agents are more exploitable at low temperatures, suggesting our agents usefully mix strategies. Final training targets are less exploitable than final networks, including in IBR, unlike what we'd expect if we used an exact BR operator, which would yield a highly exploitable deterministic policy. For full details see Appendix B.

# 6 Conclusion

We proposed a novel approach for training RL agents in Diplomacy with BRPI methods and overcoming the simultaneous moves and large combinatorial action space using our simple yet effective SBR improvement operator. We set-out a thorough analysis process for Diplomacy agents. Our methods improve over the state of the art, yielding a consistent improvement of the agent policy.

Our results provide generalisable lessons about learning in many-agent and imperfect information settings: We show that sampling-heavy best response estimation can be sufficient to drive learning in a complex many-agent domain. In fact, IBR was surprisingly effective, and we believe this is an result of the stochasticity from the SBR method. In Blotto, we showed that stochastic best response approximation was beneficial to BRPI algorithms. Our most successful approach was FPPI-2, we found this method of averaging policies to be stronger than FPPI-1's NFSP-style method.

Using RL to improve game-play in No-press Diplomacy is a prerequisite for investigating the complex mixed motives and many-player aspects of this game. Future work can now focus on questions like: (1) What is needed to achieve human-level No-Press Diplomacy AI? (2) How can the exploitability of RL agents be reduced? (3) Can we build agents that reason about the incentives of others, for example behaving in a reciprocal manner [29], or by applying opponent shaping [36]? (4) How can agents learn to use signalling actions to communicate intentions in No-Press Diplomacy? (5) Finally, how can agents handle negotiation in Press variants of the game, where communication is allowed?

## Broader Impact

We discuss the potential impact of our work, examining possible positive and negative societal impact.

**What is special about Diplomacy?** Diplomacy [19] has simple rules but high emergent complexity. It was designed to accentuate dilemmas relating to building alliances, negotiation and teamwork in the face of uncertainty about other agents. The tactical elements of Diplomacy form a difficult environment for AI algorithms: the game is played by seven players, it applies simultaneous moves, and has a very large combinatorial action space.

**What societal impact might it have?** We distinguish immediate societal impact arising from the availability of the new training algorithm, and indirect societal impact due to the future work on many-agent strategic decision making enabled or inspired by this work.

**Immediate Impact.** Our methods allow training agents in Diplomacy and other temporally extended environments where players take simultaneous actions, and the action of a player can be decomposed into multiple sub-actions, in domains that can can be simulated well, but in which learning has been difficult so far. Beyond the direct impact on Diplomacy, possible applications of our method include business, economic, and logistics domains, in as far as the scenario can be simulated sufficiently accurately. Examples include games that require a participant to control multiple units (Starcraft and Dota [119, 14] have this structure, but there are many more), controlling fleets of cars or robots [1, 93, 118] or sequential resource allocation [95, 91]. However, applications such as in business or logistics are hard to capture realistically with a simulator, so significant additional work is needed to apply this technology in real-world domains involving multi-agent learning and planning.

While Diplomacy is themed as a game of strategy where players control armies trying to gain control of provinces, it is a very abstract game - not unlike Chess or Checkers. It seems unlikely that real-world scenarios could be successfully reduced to the level of abstraction of a game like Diplomacy. In particular, our current algorithms assume a known rule set and perfect information between turns, whereas the real world would require planning algorithms that can manage uncertainty robustly.

**Future Impact.** In providing the capability of training a tactical baseline agent for Diplomacy or similar games, this work also paves the way for research into agents that are capable of forming alliances and use more advanced communication abilities, either with other machines or with humans. In Diplomacy and related games this may lead to more interesting AI partners to play with. More generally, this line of work may inspire further work on problems of cooperation. We believe that a key skill for a Diplomacy player is to ensure that, wherever possible, their pairwise interactions with other players are positive sum. AIs able to play Diplomacy at human level must be able to achieve this in spite of the incentive to unilaterally exploit trust established with other agents.

More long term, this work may pave the way towards research into agents that play the full version of the game of Diplomacy, which includes communication. In this version, communication is used to broker deals and form alliances, but also to misrepresent situations and intentions. For example, agents may learn to establish trust, but might also exploit that trust to mislead their co-players and gain the upper hand. In this sense, this work may facilitate the development of manipulative agents that use false communication as a means to achieve their goals. To mitigate this risk, we propose using games like Diplomacy to study the emergence and detection of manipulative behaviours in a sandbox — to make sure that we know how to mitigate such behaviours in real-world applications.

Overall, our work provides an algorithmic building block for finding good strategies in many-agent systems. While prior work has shown that the default behaviour of independent reinforcement learning agents can be non-cooperative [74, 36, 54], we believe research on Diplomacy could pave the way towards creating artificial agents that can successfully cooperate with others, including handling difficult questions that arise around establishing and maintaining trust and alliances.

## Acknowledgements

We thank Julia Cohen, Oliver Smith, Dario de Cesare, Victoria Langston, Tyler Liechty, Amy Merrick and Elspeth White for supporting the project. We thank Edward Hughes and David Balduzzi for their advice on the project. We thank Kestas Kuliukas for providing the dataset of human diplomacy games.

## Author Contributions

T.A., T.E., A.T., J.K., I.G. and Y.B. designed an implemented the RL algorithm. I.G., T.A., T.E., A.T., J.K., R.E., R.W. and Y.B. designed and implemented the evaluation methods. A.T., J.K., T.A., T.E., I.G. and Y.B. designed and implemented the improvements to the network architecture. T.H. and N.P. wrote the Diplomacy adjudicator. M.L. and J.P. performed the case-study on Blotto and theoretical work on FP. T.A., M.L. and Y.B. wrote the paper. S.S. and T.G. advised the project

## Funding Statement

Authors are employees of DeepMind.

## Footnotes

[1]For comparison, Chess's game tree size is $10^{123}$, it has $10^{47}$ states, and fewer than 100 legal actions per turn. Estimates for Diplomacy are based on human data; see Appendix I for details.

[2]A similar effect could be achieved with a DAgger-like procedure [100], or reservoir sampling [49].

[3]The score is 1 for a win, $\frac{1}{n}$ for $n$ players surviving at a timeout of $\sim 80$ game-years, and 0 otherwise.

[4]A similar analysis called a 'Nash League' was used to study Starcraft agents [119].

[5]We will open-source these BRPI agents and our SL agent for benchmarking.

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
