[Supplementary Material]

# Appendices

## Contents

## A Case Study: Many-Player Colonel Blotto

Our improvement operator for Diplomacy is SBR, described in Section 3.1. Diplomacy is a complex environment, where training requires significant time. To analyze the impact of applying SBR as an improvement operator, we use the much simpler normal-form game called the Colonel Blotto game [15] as an evaluation environment. We use the Blotto environment to examine several variants of best response policy iteration. We run these experiments using OpenSpiel [72] and the code is available in `examples/sbr_blotto`.

Colonel Blotto is a famous game theoretic setting proposed about a century ago. It has a very large action space, and although its rules are short and simple, it has a high emergent complexity [96]. The original game has only two players, but we describe an $n$-player variant, Blotto($n, c, f$): each player has $c$ coins to be distributed across $f$ fields. The aim is to win the most fields. Each player takes a single action simultaneously and the game ends. The action is an allocation of the player's coins across the fields: the player decides how many of its $c$ coins to put in each of the fields, choosing $c_1, c_2, \ldots, c_f$ where $\sum_{i=1}^{f} c_i = c$. For example, with $c = 10$ coins and $f = 3$ fields, one valid action is [7, 2, 1], interpreted as 7 coins on the first field, 2 on the second, and 1 on the third. A field is won by the player contributing the most coins to that field (and drawn if there is a tie for the most coins). The winner receives a reward or +1 (or a +1 is shared among all players with the most fields in the case of a tie), and the losers share a -1, and all player receive 0 in the case of a $n$-way tie.

We based our analysis on Blotto for several reasons. First, like Diplomacy, it has the property that the action space grows combinatorially as the game parameter values increase. Second, Blotto is a game where the simultaneous-move nature of the game matters, so players have to employ mixed strategies, which is a difficult case for standard RL and best response algorithms. Third, it has been used in a number of empirical studies to analyze the behavior of human play [3, 62]. Finally, Blotto is small enough that distances to various equilibria can be computed exactly, showing the effect each setting has on the empirical convergence rates.

Blotto differs to Diplomacy in several ways. The Nash equilibrium in Blotto tends to cover a substantial proportion of the strategy space, whereas in Diplomacy, many of the available actions are weak, so finding good actions is more of a needle-in-a-haystack problem. In many-agent Blotto, it is difficult for an agent to target any particular opponent with an action, whereas in Diplomacy most attacks are targeted against a specific opponent. Finally, Blotto is a single-turn (i.e. normal form) game, so value functions are not needed. Throughout this section, we use the exact game payoff wherever the value function would be needed in Diplomacy.

## A.1 Definitions and Notation Regarding Equilibria

We now provide definitions and notation regarding various forms of equilibria in games, which we use in other appendices also.

Consider an $N$-player game where players take actions in a set $\{A^i\}_{i \in \{1,\dots,N\}}$. The reward for player $i$ under a profile $\mathbf{a} = \{a_1, \dots, a_N\}$ of actions is $r^i(a_1, \dots, a_N)$.

Each player uses a policy $\{\pi_i\}_{i \in \{1,\dots,N\}}$ which is a distribution over action $A^i$.

We use the following notation:

$$r^i_\pi = \mathbb{E}_{\forall i, a^i \sim \pi^i(.)} \left[ r^i(a_1, \dots, a_N) \right]$$

$$r^i_{\pi^{-i}} = \mathbb{E}_{\forall j \neq i, a^j \sim \pi^j(.)} \left[ r^i(a_1, \dots, a_N) \right]$$

**Nash equilibrium:** A Nash equilibrium is strategy profile $(x_1, \dots, x_N)$ such that no player $i$ can improve their win rate by unilaterally deviating from their sampling distribution $x_i$ (with all other player distributions, $x_{-i}$, held fixed). Formally, a Nash equilibrium is a policy $\pi = \{\pi_1, \dots, \pi_N\}$ such that:

$$\forall i, \max_{\pi'^i} \langle \pi'^i, r^i_{\pi^{-i}} \rangle = r^i_\pi$$

$\epsilon$**-Nash:** An $\epsilon$-Nash is a strategy profile such that the most a player can improve their win rate by unilaterally deviating is $\epsilon$:

$$\mathcal{L}_{\exp_i}(\boldsymbol{x}) = \max_z \{r^i(z, x_{-i})\} - r^i(x_i, x_{-i}) \leq \epsilon \tag{1}$$

where $r^i$ is player $i$'s reward given all player strategies.

The NashConv metric, defined in [73], has a value of 0 at a Nash equilibrium and can be interpreted as a distance from Nash. It is the sum of over players of how much each player can improve their winrate with a unilateral deviation.

**Coarse Correlated Equilibrium:** The notion of a Coarse Correlated Equilibrium [86] relates to a joint strategy $\pi(a_1, \dots, a_N)$, which might not factorize into independent per-player strategies. A joint strategy $\pi(a_1, \dots, a_N)$ is a Coarse Correlated Equilibrium if for all player $i$:

$$\max_{a'_i} E_{a_1,\dots,a_N \sim \pi} \left[ r(a'_i, a_{-i}) \right] - E_{a_1,\dots,a_N \sim \pi} \left[ r(a_1, \dots, a_N) \right] \leq 0 \tag{2}$$

A joint strategy $\pi(a_1, \dots, a_N)$ is a $\epsilon$-Coarse Correlated Equilibrium if for all player $i$:

$$\max_{a'_i} E_{a_1,\dots,a_N \sim \pi} \left[ r(a'_i, a_{-i}) \right] - E_{a_1,\dots,a_N \sim \pi} \left[ r(a_1, \dots, a_N) \right] \leq \epsilon \tag{3}$$

We define a similar metric to NashCov for the empirical distance to a *coarse-correlated equilibrium* (CCE) [86], given its relationship to no-regret learning algorithms. Note, importantly, when talking about the multiplayer ($n > 2$) variants, $\pi^t_{-i}$ is always the average joint policy over all of player $i$'s opponents, rather than the player-wise marginal average policies composed into a joint policy (these correspond to the same notion of average policy for the special case of $n = 2$, but not generally).

**Definition 1** (CCEDist). *For an $n$-player normal form game with players $i \in \mathcal{N} = \{1 \cdots n\}$, reward function $r$, joint action set $\mathcal{A} = \mathcal{A}_1 \times \cdots \times \mathcal{A}_n$, let $a_{i \to *}$ be the joint action where player $i$ replaces their action with $a_i^*$; given a correlation device (distribution over joint actions) $\mu$,*

$$\text{CCEDIST}(\mu) = \sum_{i \in \mathcal{N}} \max(0, \max_{a_i^* \in \mathcal{A}_i} \mathbb{E}_{a \sim \mu}[r^i(a_{i \to *}, a_{-i}) - r^i(a)]).$$

### A.1.1 Relating $\epsilon$-Coarse Correlated Equilibria, Regret, and CCEDist

In this subsection, we formally clarify the relationships and terminology between similar but slightly different concepts[6].

Several iterative algorithms playing an $n$-player game produce a sequence of policies $\pi^t = (\pi_1^t, \cdots, \pi_n^t)$ over iterations $t \in \{1, 2, \cdots, T\}$. Each individual player $i$ is said to have (external) **regret**,

$$R_i^T = \max_{a_i'} \sum_{t=1}^T r^i(a_i', \pi_{-i}^t) - \sum_{t=1}^T r^i(\pi^t),$$

where $r^i$ is player $i$'s *expected* reward (over potentially stochastic policies $\pi^t$). Define the **empirical average joint policy** $\mu$ to be the a uniform distribution over $\{\pi^t \mid 1 \le t \le T\}$. Denote $\delta_i(\mu, a_i')$ to be the incentive for player $i$ to deviate from playing $\mu$ to instead always choosing $a_i'$ over all their iterations, i.e. how much the player would have gained (or lost) by applying this deviation. Now, define player $i$'s incentive to deviate to their best response action, $\epsilon_i$, and notice:

$$\epsilon_i = \max_{a_i'} \delta_i(\mu, a_i') = \mathbb{E}_{\pi^t \sim \mu}[R_i^t] = \frac{R_i^T}{T},$$

where, in an $\epsilon$-CCE, $\epsilon = \max_i \epsilon_i$ (Equation 3). Also, a CCE is achieved when $\epsilon \le 0$ (Equation 2). Finally, CCEDist is a different, but very related, metric from $\epsilon$. Instead, it sums over the players and discards negative values from the sum, so $\text{CCEDIST}(\mu) = \sum_i \max(0, \epsilon_i)$.

## A.2 A Generalization of Best Response Policy Iteration

Algorithm 3 presents the general family of response-style algorithms that we consider (covering our approaches in Section 3.2).

---

**Algorithm 3** A Generalization of BRPI

---

**Require:** arbitrary initial policy $\pi^0$, total steps $T$
    **for** time step $t \in \{1, 2, \dots, T\}$ **do**
        **for** player $i \in \{1, \cdots, n\}$ **do**
            (response, values) $\leftarrow$ COMPUTERESPONSE$(i, \pi_{-i}^{t-1})$
            $\pi_i^t \leftarrow$ UPDATEPOLICY$(i,$ response, values$)$
    **return** $\pi^T$

---

Several algorithms fit into this general framework. For example, the simplest is tabular iterated best response (IBR), where the UPDATEPOLICY simply overwrites the policy with a best response policy. Classical fictitious play (FP) is obtained when COMPUTERESPONSE returns a best response and UPDATEPOLICY updates the policy to be the average of all best responses seen up to time $t$. Stochastic fictitious play (SFP) [38] is obtained when the best response policy is defined as a softmax over action values rather than the argmax, i.e. returning a policy

$$\pi_i(a) = \text{SOFTMAX}_\lambda(r^i)(a) = \frac{\exp(\lambda r^i(a))}{\sum_a \exp(\lambda r^i(a))},$$

where $r^i$ is a vector of expected reward for each action. Exploitability Descent [80] defines UPDATEPOLICY based on gradient descent. We describe several versions of the algorithm used in Diplomacy below. A spectrum of game-theoretic algorithms is described in [73].

We run several experiments on different game instances. The number of actions and size of each is listed in Table 2.

## A.3 Warm-up: Fictitious Play versus Iterated Best Response (IBR)

We first analyze convergence properties of various forms of BRPI, highlighting the difference between IBR and FP approaches.

| $n$ | $c$ | $f$ | Number of actions per player ($\lvert A_i \rvert$) | Size of matrix / tensor ($= \lvert A_i \rvert^n$) |
|---|---|---|---|---|
| 2 | 10 | 3 | 66 | 4356 |
| 2 | 30 | 3 | 496 | 246016 |
| 2 | 15 | 4 | 816 | 665856 |
| 2 | 10 | 5 | 1001 | 1002001 |
| 2 | 10 | 6 | 3003 | 9018009 |
| 3 | 10 | 3 | 66 | 287496 |
| 4 | 8 | 3 | 45 | 4100625 |
| 5 | 6 | 3 | 28 | 17210368 |

Table 2: Blotto Game Sizes

Figure 5: Empirical Convergence in (a) NashConv in Blotto(2, 10, 3), (b) NashConv in Blotto(3, 10, 3) using marginalized policies, (c) NashConv in Blotto(3, 10, 3), (d) NashConv in Blotto(3, 10, 3) over 10 runs of FP and SFP with random starts, (e) CCEDist in Blotto(3, 10, 3) over 10 runs of FP and SFP with random starts, and (f) CCEDist in Blotto(3, 10, 3).

Figure 5(a) shows the convergence rates to approximate Nash equilibria of fictitious play and iterated best response in Blotto(2, 10, 3). We observe that FP is reducing NashConv over time while IBR is not. In fact, IBR remains fully exploitable on every iteration. It may be cycling around new best responses, but every best response is individually fully exploitable: for every action $[x, y, z]$ there exists a response of the form $[x + 1, y + 1, z - 2]$ which beats it, so playing deterministically in Blotto is always exploitable, which demonstrates the importance of using a stochastic policy.

The convergence graphs for FP and IBR look similar with $n = 3$ players (Figure 5(b), despite both being known to not converge generally, see [58] for a counter-example). One problem with IBR in this case is that it places its entire weight on the newest best response. FP mitigates this effectively by averaging, i.e. only moving toward this policy with a step size of $\alpha_t \approx \frac{1}{t}$. One question, then, is whether having a stochastic operator could improve IBR's convergence rate. We see that indeed MaxEnt IBR– which returns a mixed best response choosing uniformly over all the tied best response actions– does find policies that are not fully-exploitable; however, the NashConv is still non-convergent and not decreasing over time.

Finally, we consider the empirical convergence rates of Stochastic FP (SFP). Figures 5(a) and (b) show a basic version of FP and SFP that marginalize the policies so that $\pi^t_{-i}$ and $\pi^t$ are products of individual $\pi^t_i$ over players $i \in \{1, \cdots n\}$. The plateaus in Figure 5(a) occur due to the fact that the introduction of the softmax induces convergence towards Quantal Response Equilibria (QRE) [82, 51, 39] rather than Nash equilibria and can be interpreted as entropy-regularizing the reward [92]. There is a clear trend of SFP reaching its plateau, at lower NashConv, significantly earlier than FP in both 2 and 3-player cases. Also, the choice of lambda determines the lower-bound

on NashConv. However, at a higher values of $\lambda = 100$, there appears to be long-term instability that was not observed in the 2-player zero-sum game.

Inspired by foundations of fictitious play [88, 40], we now adopt FP to correlate $\pi^t$ and $\pi^t_{-i}$ by storing a single distribution over past *joint* policies $\pi^t$. Fig 5(c) is then a rerun of Fig 5(b) with this one change. Observe that in Figure 5 (a), (b), and (c), as before the choice of the $\lambda$ has a strong effect on the convergence curve, improving with high $\lambda$, but with the a similar long-term instability at $\lambda = 100$. As SFP is not guaranteed to reduce NashConv in this 3-player game, we also show the empirical convergence rates of CCEDist in Figure 5(f); here we notice that the value seems to stay close to its lower bound. To investigate further, we ran several instances of FP and SFP from random starting points (i.e. initial policy generated by normalizing uniformly drawn random numbers); results are shown in Fig 5(d) and (e), using the same set of seeds.

Random initial starting points clearly affect the resulting curves, suggesting that learning in many-player settings might be particularly unstable. Reducing NashConv in 3-player Blotto seems difficult for both FP and SFP, more so for SFP. However, in two of the ten runs, FP plateaus early at a high value. The same occurs for FP using CCEDist, however all of the SFP runs remain near their expected plateau; specifically, we do not observe any long-term instabilities present in the NashConv metric. In addition, SFP is Hannan-consistent [40] and can be interpreted as a probabilistic best response operator. We follow-up on convergence properties of SFP in Appendix H.

## A.4 Fictitious Play using SBR

In this subsection, we refer to FP+SBR$(B, C)$ as the instance of algorithm 3 that averages the policies like fictitious play but uses a Sampled Best Response operator as described in Algorithm 1, using $B$ base profiles and $C$ candidates, where the base profile sampling policy is simply $\pi^t$ and the candidate sampling policy is uniform over all actions.

### A.4.1 Trade-offs and Scaling

Figure 6: Convergence of Fictitious Play versus FP+SBR(10,50) by elapsed time in 2-player Blotto with increasing action space sizes where (a) > (b) > (c) > (d).

Figure 6 shows several convergence graphs of different 2-player Blotto games with increasing action sizes using elapsed time as the x-axis. The first observation is that, in all cases, FP+SBR computes a policy in a matter of milliseconds, thousands of times earlier than FP's first point. Secondly, it

appears that as the action space the game grows, the point by which SBR achieves a specific value and when FP achieves the same value gets further apart: that is, SBR is getting a result with some quality sooner than FP. For example, in Blotto(2, 10, 6), SBR can achieve the an approximation accuracy in 1 second which takes FP over three hours to achieve. To quantify the trade-offs, we compute a factor of elapsed time to reach the value by FP divided by elapsed time taken to reach NashConv $\approx 0.2$ by FP+SBR(10,50). These values are 1203, 1421, 2400, and 2834 for the four games by increasing size. This trade-off is only true up to some threshold accuracy; the benefit of approximation from sampling from SBR leads to plateaus long-term and is eventually surpassed by FP. However, for large enough games even a single full iteration of FP is not feasible.

Figure 7: Convergence of Fictitious Play versus FP+SBR(10,25) by elapsed time in (a) Blotto(3, 10, 3) (b) Blotto(4, 6, 3), and (c) Blotto(5, 6, 3).

The trade-offs are similar in $(n > 2)$-player games. Figure 7 shows three games of increasing size with reduced action sizes to ensure the game was not too large so joint policies could still fit into memory. There is a similar trade-off in the case of 3-player, where it is clear that FP catches up. In Blotto(4,8,3), even $> 25000$ seconds was not enough for the convergence of FP to catch-up, and took $> 10000$ seconds for Blotto(5,6,3). In each case, FP+SBR took at most 1 second to reach the CCEDist value at the catch-up point.

### A.4.2 Effects of choices of $B$ and $C$

Figure 8: Convergence rates of FP+SBR$(B, C)$ for various settings of $B$ and $C$. The first row uses the game of Blotto(3, 10, 3), second row Blotto(4, 8, 3), and third row Blotto(5, 6, 3). The columns represent $B = 4$, $B = 32$, and $B = 64$, respectively from left to right.

Figure 8 shows the effect of varying $B$ and $C$ in FP+SBR$(B, C)$. Low values of $B$ clearly lead to early plateaus further from equilibrium (graphs for $B < 4$ look similar). At low number of base profiles, it seems that there is a region of low number of candidate samples (2-4) that works best for which plateau is reached, presumably because the estimated maximum over a crude estimate of the expected value has smaller error. As $B$ increases, the distance to equilibria becomes noticeably smaller and sampling more candidates works significantly better than at low $B$.

In the two largest games, FP+SBR(64, 64) was able to reach a CCEDist $\leq 0.3$ while fictitious play was still more than three times further from equilibrium after six hours of running time.

### A.5 BRPI Convergence and Approximation Quality

FP+SBR is an idealized version of the algorithm that explicit performs policy averaging identical to the outer loop of fictitious play: only the best response step is replaced by a stochastic operator.

We now analyze an algorithm that is closer to emulating BRPI as described in the main paper. Due to stochasticity the policy, the policy trained by BRPI at iteration $t$ for player $i$ can be described as the empirical (joint) distribution:

$$\pi^t = \frac{1}{N} \sum_{n=1}^{N} \mathbf{1}(a), \text{ where } a \sim \text{SBR}(\pi_b^t, \pi_c^t, B, C),$$

where $\mathbf{1}(a)$ is the deterministic joint policy that chooses joint action $a$ with probability 1, SBR is the stochastic argmax operator defined in Algorithm 1, and $\{\pi_b^t, \pi_c^t\}$ are the base profile and candidate sampling policies which are generally functions of $(\pi^0, \pi^t, \cdots, \pi^{t-1})$.

The average of the operator over $N$ samples models the best possible fit to dataset of $N$ samples

#### A.5.1 Effects of choices of $B$ and $C$

To start, in order to compare to the idealized form, we show similar graphs using settings which most closely match FP+SBR: $\pi_b^t$ uniformly $t \sim \text{UNIF}(\{0, \cdots, t-1\}$ and then samples a base profile from $\pi^t$, and $\pi_c$ samples from the initial policy where all players play each action uniformly at random.

Figure 9 show effects of varying $B$ and $C$ over the games. Note that convergence to the plateau is much faster than FP+SBR, presumably because of the $N$ samples per iteration rather than folding one sample into the average. Like with FP+SBR, the value of $B$ has a strong effect on the plateau that is reached, and this value is separated by the choice of $C$. Unlike FP+SBR the value of $C$ has a different effect: higher $C$ is generally better at lower values of $B$. This could be due to the fact that, in BRPI, the only way the algorithm can represent a stochastic policy is via the $N$ samples, whereas FP+SBR computes the average policy exactly; the error in the max over a crude expectation may be less critical than having a granularity of a fixed limit of $N$ samples.

This is an encouraging result as it shows that a mixed policy can be trained through multiple samples from a stochastic best response operator *on each iteration*, rather than computing the average policy explicit. However, this comes at the extra cost of remembering all the past policies; in large games, this can be done by saving checkpoints of the network periodically and querying them as necessary.

#### A.5.2 Varying the Candidate Sampling Policy

Most of the runs look similar to the previous subsection (convergence plateau is mostly reached within $60 - 100$ seconds), so to demonstrate the effect of the various candidate sampling policies, we instead show the CCEDist reached after running for a long time ($> 50000$ seconds). This roughly captures the asymptotic value of each method, rather than its convergence rate.

Figure 10 shows the long-term CCEDist reached by BRPI at $B = 2$ and $C = 16$ for various choices of the candidate sampling schemes. There is a clearly best choice of using uniformly sampled past policy to choose candidates followed by the mixtures: initial + uniform, and initial + latest.

Figure 9: Convergence rates of $\text{BRPI}(\pi_b, \pi_c, B, C)$ for various settings of $B$ and $C$ using a uniform iteration for $\pi_b$ and uniform random action for $\pi_c$, and $N = 1000$. The first row uses the game of Blotto(3, 10, 3), second row Blotto(4, 8, 3), and third row Blotto(5, 6, 3). The columns represent $B = 1$, $B = 4$, and $B = 64$, respectively from left to right.

CCEDist reached by BRPI with $\pi_b$ sampling from a uniform past policy and $B = 2$, $C = 16$

Figure 10: Long-term CCEDist reached by $\text{BRPI}(2, 16)$ with $\pi_b$ choosing a uniform past policy in Blotto(3, 10, 3) (left), Blotto(4, 8, 3) (middle), and Blotto(5, 6, 3) (right).

### A.5.3 Iterated Sampled Best Response

We now consider the case where the base sampling policy is the last policy: $\pi_b^t = \pi^{t-1}$. Figure 11 shows the long-term CCEDist reached by BRPI at $B = 2$ and $C = 16$ for various choices of the candidate sampling schemes.

In this case, there is no clear winner in all cases, but initial + latest seems to be a safe choice among these five sampling schemes. Despite the plateau values being generally higher than $\pi_b$ sampling from a uniform past policy across the candidate sampling schemes (note the y-axis scale differs

Figure 11: Long-term CCEDist reached by BRPI$(2, 16)$ with with $\pi_b^t = \pi^{t-1}$ in Blotto(3, 10, 3) (left), Blotto(4, 8, 3) (middle), and Blotto(5, 6, 3) (right).

between the two plots), the values under the initial + latest sampling policy are matched in two of the three games.

Though using the last policy for $\pi_b$ yields generally higher final plateaus in CCEDist, the fact that these are comparable to those achieved by FP+SBR stands in stark contrast to when we use an exact best response operator in A.3; in that case, Iterated Best Response makes no progress. This shows that using a Sampled Best Response in the place of an exact one can dramatically improve the behaviour of Iterated Best Response.

# B  Additional Results

## B.1  Behavioural analysis

We present some descriptive statistics of the behaviour exhibited by the different networks in self-play games, and by human players in the datasets. We examine the move-phase actions of agents, investigating the tendency of agents to support another power's unit to move or hold, which we refer to as "cross power support". We also examine the success rates, which are defined by whether the other power made a corresponding move for that unit (respectively, either the target move or a non-moving order). Figure 12a compares the proportion of actions that are cross power support across different agents, and their success (for both holding and moving). The results indicate the BRPI agents have a substantially reduced rate of cross power hold support, and the BRPI agents have a substantially increased rate of cross-power move support. The A2C agent attempts both types of support less often but succeeds a higher proportion of the time.

This analysis is related but different to the cross-support analysis in [90], which considers cross-power supports as a proportion of supports, and rather than looking at "success" as we've defined it for support orders, they measure "effectiveness", defined in terms of whether the support made a difference for the success of the move order or defence being supported.

We also examine the propensity of agents to *intrude on* other agents, defined as one of the following:

- a move order (including via convoy) into a territory on which another agent has a unit

- a move order (including via convoy) into a supply center owned by another agent (or remaining/holding in another agent's supply center during fall turns)

- successfully supporting/convoying a move falling in the categories above

(a) Comparison of the cross-power support behaviours of different networks.

(b) Comparison of peace correlations between different networks.

Figure 12: Descriptive behavioural statistics of the different networks, as well as human play in the datasets.

We define two powers to be *at conflict* in a moves phase[7] if either one intrudes upon the other, and to be *distant* if neither one has the option of doing that. Then we define the *peace proportion* of a network to be the proportion, among instances in which powers are non-distant, that they're not at conflict; and the *peace correlation* to be the correlation, among those same instances, between conflict in the current moves phase and conflict in the next moves phase. In Figure 12b we compare these statistics across our different networks. These results indicate that BRPI reduces the peace proportion while maintaining the peace correlation, while A2C brings down both the peace proportion and the peace correlation significantly.

We used sampling temperature $0.1$ for these agents, and considered only the first 10 years of each game in order to make the comparisons more like-for-like, particularly to human games. We used 1000 self-play games for each network; results for IBR, FPPI-1, and FPPI-2 were combined from results for the final networks from 5 training runs. In addition we included the evaluation sets from both our press and no-press datasets, excluding games that end within 5 years and games where no player achieved 7 supply centers.

## B.2  Exploitability

In this section we give more details on our experiments on the exploitability of our networks. We use for two different exploiters for each agent we exploit, both of which are based on the Sampled Best Response operator, using a small number of samples from the policy as the base profiles to respond to.

Firstly, we use a few shot exploiter. Apart from using base profiles from the policy being exploited, but otherwise is independent from it – the value function for SBR is taken from an independent

BRPI run (the same for all networks exploited), and the candidates from the human imitation policy; SBR($\pi^c = \pi^{\text{SL}}$, $\pi^b = \pi$, $v = V^{\text{RL}}$). This has the advantage of being the most comparable between different policies; the exploits found are not influenced by the strength of the network's value function, or by the candidates they provide to SBR. This measure should still be used with care; it is possible for an agent to achieve low few-shot exploitability without being strong at the game, for example by all agents playing pre-specified moves, and uniting to defeat any deviator.

The other exploiter shown is the best found for each policy. For policies from the end of BRPI training, this is SBR($\pi^c = \pi^{\text{SL}} + \pi$, $\pi^b = \pi$, $v = V^\pi$), which uses a mixture of candidates from the exploitee and supervised learning, and the exploitee's value function. For $\pi^{SL}$, we instead use SBR($\pi^c = \pi^{\text{SL}} + \pi^{\text{RL}}$, $\pi^b = \pi$, $v = V^{\text{RL}}$), where $\pi^{\text{RL}}$ and $v = V^{\text{RL}}$ are from a BRPI run. This is because the value function learned from human data is not correct for $\pi^{SL}$, and leads to weak exploits. Still, we find that the best exploit found for the supervised learning agent is much closer to those found for BRPI agents than for few-shot exploits. This highlights a limitation of this method – the BRPI exploiters are much better tuned to exploiting these networks than the exploiter used for $\pi^{SL}$.

Figure 13 shows the winrates achieved by each of these exploiters playing with 6 copies of a network, for our supervised learning agent and the final agent from one run of each BRPI setting. All these networks are least exploitable at $t = 0.5$; this appears to balance the better strategies typically seen at lower temperatures with the mixing needed to be relatively unexploitable. In the few-shot regime, IBR and FPPI-2 produce agents which are less exploitable than the supervised learning agent; for the best exploiters found, the picture is less clear. This may be because we do not have as good a value function for games with $\pi^{SL}$ as we do for the other policies.

Tables 3 and 4 shows lower bounds on the exploitability of the final training targets from the three RL runs, again with few-shot and best exploiters. This gives a lower bound for exploitability which is less than for the networks these targets are improving on. This is particularly interesting for the training target for IBR – which consists of a single iteration of Sampled Best Response. This is in contrast with what we would see with an exact best response, which would be a highly exploitable deterministic policy.

| | 1 profile | 2 profiles | 4 profiles |
|---|---|---|---|
| SL | $0.170 \pm 0.010$ | $0.216 \pm 0.010$ | $0.261 \pm 0.011$ |
| IBR | $0.045 \pm 0.009$ | $0.123 \pm 0.010$ | $0.185 \pm 0.012$ |
| FPPI-1 | $0.057 \pm 0.009$ | $0.130 \pm 0.011$ | $0.182 \pm 0.012$ |
| FPPI-2 | $0.043 \pm 0.008$ | $0.115 \pm 0.010$ | $0.167 \pm 0.010$ |
| Target(IBR) | $0.000 \pm 0.015$ | $0.070 \pm 0.018$ | $0.149 \pm 0.024$ |
| Target(FPPI-1) | $-0.018 \pm 0.011$ | $0.055 \pm 0.014$ | $0.112 \pm 0.019$ |
| Target(FPPI-2) | $0.002 \pm 0.010$ | $0.070 \pm 0.013$ | $0.158 \pm 0.018$ |

Table 3: Few-shot exploitability of final networks training targets with different numbers of base profiles. Networks are shown at the temperature with the highest lower bound on exploitability.

| | 1 profile | 2 profiles | 4 profiles |
|---|---|---|---|
| SL | $0.170 \pm 0.010$ | $0.216 \pm 0.010$ | $0.261 \pm 0.011$ |
| IBR | $0.156 \pm 0.010$ | $0.250 \pm 0.012$ | $0.325 \pm 0.012$ |
| FPPI-1 | $0.129 \pm 0.010$ | $0.201 \pm 0.011$ | $0.275 \pm 0.013$ |
| FPPI-2 | $0.123 \pm 0.010$ | $0.218 \pm 0.011$ | $0.266 \pm 0.011$ |
| Target(IBR) | $0.056 \pm 0.016$ | $0.115 \pm 0.019$ | $0.193 \pm 0.023$ |
| Target(FPPI-1) | $0.042 \pm 0.013$ | $0.125 \pm 0.016$ | $0.240 \pm 0.022$ |
| Target(FPPI-2) | $0.064 \pm 0.013$ | $0.151 \pm 0.016$ | $0.230 \pm 0.021$ |

Table 4: Best lower bound on exploitability of final networks and training targets with different numbers of base profiles. Networks are shown at the temperature with the highest lower bound on exploitability.

All the exploiting agents here use 64 candidates for SBR at each temperature in $(0.1, 0.25, 0.5, 1.0)$.

Figure 13: Exploiter winrates of imitation and final BRPI networks. Left column shows the exploits achieved by few shot exploiters, right column the best exploiters found.

### B.3  Head to head comparisons

Here, we add to Table 1 additional comparisons where we run a test-time improvement step on our initial and final networks. These steps use a more expensive version of SBR than we use in training; we sample 64 candidates from the network at each of four temperatures $(0.1, 0.25, 0.5, 1.0)$, and also sample the base profiles at temperature $0.1$. We only compare these training targets to the networks and training targets for other runs – comparing a training target to the network from the same run would be similar to the exploits trained in Appendix B.2.

For the final networks, these improvement steps perform well against the final networks (from their own algorithm and other algorithms). For the initial network, the resulting policy loses to all agents

except the imitation network it is improving on; we hypothesise that this is a result of the value function, which is trained on the human dataset and so may be inaccurate for network games.

We also add the agent produced by our implementation of A2C, as described in E.

| | SL [90] | A2C [90] | SL (ours) | A2C (ours) | FPPI-1 net | IBR net | FPPI-2 net | SBR-SL | SBR-IBR | SBR-FPPI-2 |
|---|---|---|---|---|---|---|---|---|---|---|
| SL [90] | 14.3% | 7.9% | 16.3% | 8.3% | 3.1% | 1.9% | 1.4% | 30.1% | 3.1% | 3.7% |
| A2C [90] | 15.5% | 14.3% | 15.5% | 16.6% | 3.0% | 2.3% | 1.3% | 43.6% | 4.7% | 4.4% |
| SL (ours) | 12.4% | 8.5% | 14.3% | 11.1% | 3.9% | 2.3% | 2.0% | 18.2% | 2.9% | 1.9% |
| A2C | 13.9% | 3.6% | 17.7% | 14.3% | 3.2% | 2.2% | 2.1% | 30.7% | 3.2% | 3.9% |
| FPPI-1 net | 30.1% | 27.3% | 28.9% | 33.0% | 13.4% | 7.0% | 6.1% | 52.9% | 6.9% | 5.0% |
| IBR net | 23.3% | 26.8% | 24.1% | 29.2% | 12.6% | 13.6% | 12.6% | 53.7% | 8.0% | 10.4% |
| FPPI-2 net | 20.4% | 26.4% | 24.7% | 28.2% | 13.5% | 15.3% | 13.5% | 56.9% | 9.6% | 9.4% |
| SBR-SL | 14.0% | 4.5% | 21.8% | 12.9% | 1.9% | 0.8% | 0.7% | 14.3% | 0.8% | 0.6% |
| SBR-IBR | 24.5% | 22.5% | 31.8% | 29.7% | 24.4% | 39.8% | 31.3% | 47.3% | 15.7% | 18.3% |
| SBR-FP-2 | 23.3% | 23.7% | 34.6% | 27.4% | 29.9% | 28.8% | 28.1% | 49.2% | 15.9% | 16.6% |

Table 5: Matches between different algorithms. Winrates for 1 row player vs 6 column players

In Table 6, we give comparisons for the final networks of agents trained using the design space of BRPI specified in 3.2. The notation for candidate policies and base profiles is $\pi_0$ for the imitation network we start training from, $\pi_{t-1}$ is the policy from the previous iteration, and $\mu_{t-1}$ is the average of the policies from previous iterations. $V_{t-1}$ is the value function from the previous iteration, and $V_{t-1}^{\mu}$ is the average value function from the previous iterations. When $\mu_{t-1}$ and $V_{t-1}^{\mu}$ are used, the sampling is coupled; the value function is from the same network used for the candidates and/or base profiles.

The fourth column of Table 6 records which kind of BRPI method each is. Methods that use the latest policy only for base profiles are IBR methods, uniformly sampled base profiles are FP methods. FP methods that use the latest networks for the value function or for candidate sampling do not recreate the historical best responses, so are FPPI-2 methods. The remaining methods are FPPI-1 methods. The asterisks mark the examples of IBR, FPPI-1 and FPPI-2 selected for deeper analysis in section 5; these were chosen based on results of experiments during development that indicated that including candidates from the imitation policy was helpful. In particular, they were not selected based on the outcome of the training runs presented in this work.

We find that against the population of final networks from all BRPI runs (1 vs 6 BRPI), IBR and FPPI-2 do better than FPPI-1. For candidate selection, using candidates from the latest and initial networks performs best. For beating the DipNet baseline, we find that using candidates from the imitation policy improves performance substantially. This may be because our policies are regularised towards the style of play of these agents, and so remain more able to play in populations consisting of these agents.

### B.4 Effect of BRPI hyperparameters

In Tables 7 and 8 we investigate the effect of different BRPI settings on the outcome of policy iteration, using the IBR setting. We see that BRPI is quite resilient to the choice of these hyperparameters.

## C  Imitation Learning and Neural Network Architecture

We fully describe the architecture of the neural network we use for approximating policy and value functions, including hyperparameter settings. The architecture is illustrated in Figure 14.

Our network outputs policy logits for each unit on the board controlled by the current player $p$, as well as a value estimate for $p$. It takes as inputs:

| $\pi_c$ | $\pi_b$ | $V$ | BRPI type | 1 vs 6 BRPI | 1 BRPI vs 6 | 1 vs 6 DipNet | 1 DipNet vs 6 |
|---|---|---|---|---|---|---|---|
| $\pi_0$ | $\pi_{t-1}$ | $V_{t-1}$ | IBR | 9.6% | 16.1% | 24.7% | 4.0% |
| $\pi_0$ | $\mu_{t-1}$ | $V_{t-1}$ | FPPI-2 | 9.8% | 17.7% | 24.4% | 3.7% |
| $\pi_0$ | $\mu_{t-1}$ | $V_{t-1}^{\mu}$ | FPPI-1 | 8.5% | 17.1% | 27.3% | 3.3% |
| $\mu_{t-1}$ | $\pi_{t-1}$ | $V_{t-1}$ | IBR | 15.4% | 10.9% | 25.9% | 1.3% |
| $\mu_{t-1}$ | $\mu_{t-1}$ | $V_{t-1}$ | FPPI-2 | 14.0% | 12.6% | 22.3% | 1.7% |
| $\mu_{t-1}$ | $\mu_{t-1}$ | $V_{t-1}^{\mu}$ | FPPI-1 | 9.3% | 15.6% | 24.4% | 2.3% |
| $\mu_{t-1} + \pi_0$ | $\pi_{t-1}$ | $V_{t-1}$ | IBR | 14.3% | 12.8% | 25.5% | 2.0% |
| $\mu_{t-1} + \pi_0$ | $\mu_{t-1}$ | $V_{t-1}$ | FPPI-2 | 13.8% | 12.6% | 26.1% | 1.7% |
| $\mu_{t-1} + \pi_0$ | $\mu_{t-1}$ | $V_{t-1}^{\mu}$ | FPPI-1* | 9.3% | 17.2% | 26.4% | 2.3% |
| $\pi_{t-1}$ | $\pi_{t-1}$ | $V_{t-1}$ | IBR | 16.0% | 7.5% | 14.3% | 0.5% |
| $\pi_{t-1}$ | $\mu_{t-1}$ | $V_{t-1}$ | FPPI-2 | 17.4% | 6.9% | 13.2% | 0.7% |
| $\pi_{t-1}$ | $\mu_{t-1}$ | $V_{t-1}^{\mu}$ | FPPI-2 | 9.7% | 15.5% | 13.6% | 3.5% |
| $\pi_{t-1} + \pi_0$ | $\pi_{t-1}$ | $V_{t-1}$ | IBR* | 16.4% | 9.8% | 20.7% | 1.8% |
| $\pi_{t-1} + \pi_0$ | $\mu_{t-1}$ | $V_{t-1}$ | FPPI-2* | 17.6% | 8.2% | 19.4% | 0.8% |
| $\pi_{t-1} + \pi_0$ | $\mu_{t-1}$ | $V_{t-1}^{\mu}$ | FPPI-2 | 12.5% | 13.2% | 23.9% | 1.8% |
| $\pi_0$ | mean | mean | - | 9.3% | 17.0% | 25.5% | 3.7% |
| $\mu_{t-1}$ | mean | mean | - | 12.9% | 13.0% | 24.2% | 1.8% |
| $\mu_{t-1} + \pi_0$ | mean | mean | - | 12.5% | 14.2% | 26.0% | 2.0% |
| $\pi_{t-1}$ | mean | mean | - | 14.4% | 10.0% | 13.7% | 1.6% |
| $\pi_{t-1} + \pi_0$ | mean | mean | - | 15.5% | 10.4% | 21.4% | 1.5% |
| mean | $\pi_{t-1}$ | $V_{t-1}$ | - | 14.3% | 11.4% | 22.2% | 1.9% |
| mean | $\mu_{t-1}$ | $V_{t-1}^{\mu}$ | - | 9.9% | 15.7% | 23.1% | 2.6% |
| mean | $\mu_{t-1}$ | $V_{t-1}$ | - | 14.5% | 11.6% | 21.1% | 1.7% |

Table 6: Performance of different BRPI variants against BRPI and DipNet agents. The scores are all for the 1 agent. All results are accurate to $0.5\%$ within a confidence interval of $95\%$

| Agent | (C=4, B=2) | (C=8, B=2) | (C=16, B=2) | SL [90] |
|---|---|---|---|---|
| (C=4, B=2) | 13.2% | 10.7% | 9.3% | 21.2% |
| (C=8, B=2) | 12.4% | 12.0% | 12.1% | 21.1% |
| (C=16, B=2) | 8.0% | 10.9% | 13.3% | 23.2% |
| SL [90] | 2.4% | 2.9% | 1.6% | 14.3% |

Table 7: Performance of final networks for BRPI with various numbers of candidates against each other and against DipNet, averaged across 5 runs of each BRPI setting. The scores are all for 1 row agent against 6 column agents. All results are accurate to $0.5\%$ within a confidence interval of $95\%$.

- $x_b$, a representation of the current state of the board, encoded with the same 35 board features per area [8] used in DipNet
- $x_m$, the state of the board during the last moves phase, encoded the same way
- $x_o$, the orders issued since that phase
- $s$, the current *Season*
- $p$, the current power
- $x_d$, the current build numbers (i.e. the difference between the number of supply centres and units for each power)

Similar to DipNet, we first produce a representation of previous gameplay. This incorporates learned embeddings $e_o(x_o)$, $e_s(s)$, and $e_p(p)$ applied to the recent orders, the season, and the power. The recent orders embeddings are summed in each area, producing $\tilde{e}_o(x_o)$. We begin by concatenating

| Agent | (C=16, B=1) | (C=16, B=2) | (C=16, B=4) | SL [90] |
|---|---|---|---|---|
| (C=16, B=1) | 13.5% | 13.0% | 11.1% | 19.3% |
| (C=16, B=2) | 14.0% | 13.3% | 11.8% | 23.2% |
| (C=16, B=4) | 13.0% | 13.5% | 11.7% | 17.9% |
| SL [90] | 1.6% | 1.6% | 2.2% | 14.3% |

Table 8: Performance of final networks for BRPI with various numbers of base profiles against each other and against DipNet, averaged across 5 runs of each BRPI setting. The scores are all for 1 row agent against 6 column agents. All results are accurate to $0.5\%$ within a confidence interval of $95\%$.

Figure 14: The neural network architecture for producing actions.

$x_m$ and $\tilde{e}_o(x_o)$ to produce $\tilde{x}_m$. Then, we concatenate $x_d$ and $e_s(s)$ to each of $x_b$ and $\tilde{x}_m$ to produce $\bar{x}_b = [x_b, x_d, e_s(s)]$ and $\bar{x}_m = [x_m, e_o(x_o), x_d, e_s(s)]$. (DipNet uses hardcoded "alliance features" in place of $\tilde{x}_m$, and leaves out $x_d$.)

Next, we process each of $\bar{x}_b$ and $\bar{x}_m$ with identical, but independent stacks of 12 Graph Neural Networks (GNNs) [12] linked (apart from the first layer) by residual connections. In particular, each residual GNN computes $\bar{x}^{\ell+1} = \bar{x}^\ell + \text{ReLU}(\text{BatchNorm}([\hat{\bar{x}}^\ell, A \cdot \hat{\bar{x}}^\ell]))$, where $\hat{\bar{x}}^\ell_{n,j} = \sum_i \bar{x}^\ell_{n,i} w^\ell_{n,i,j}$, with $\ell$ indexing layers in the stack, $n$ the areas, $A$ the normalized adjacency matrix of the Diplomacy board, and $\bar{x}$ being a stand-in for either $\bar{x}_b$ or $\bar{x}_m$. After concatenating $e_p(p)$ to the resulting embeddings, we use 3 additional GNN layers with residual connections to construct $x^{s,p}_b$ and $x^{s,p}_m$, which we concatenate to form our final state encoding $x^{s,p} = [x^{s,p}_b, x^{s,p}_m]$. Note that, although DipNet describes their encoders as Graph Convolutional Networks, their GNNs are not convolutional over nodes, and weights are not tied between GNNs – this produced better results, and we followed suit.

The value outputs are computed from the state encodings $x^{s,p}$ by averaging them across $p$ and $s$, and then applying a ReLU MLP with a single hidden layer to get value logits.

Like DipNet, we construct order lists from our state encoding $x^{s,p}$ considering one by one the provinces $n(1), \ldots, n(k)$ requiring an order from $p$ according to a fixed order. Unlike DipNet, we use a Relational Order Decoder (ROD). Our ROD module is a stack of 4 GNNs with residual connections that, when considering the $k$-th province, takes as input the concatenation of $x^{s,p}_{n(k)}$ and $z_{n(1),\ldots,n(k-1)}$. Precisely, $x^{s,p}_{n(k)}$ is a masked version of $x^{s,p}$ where all province representations except $n(k)$ are zeroed out, and $z_{n(1),\ldots,n(k-1)}$ contains embeddings of the orders already on the list scattered to the slots corresponding to the provinces they referred to: $n(1), \ldots, n(k-1)$. The output of the ROD corresponding to province $n(k)$ is then mapped to action logits through a linear layer

with no bias $w$. Similarly to DipNet, after sampling, the row of $w$ corresponding to the order selected is used to fill in the $n(k)$-th row of $z_{n(1),...,n(k-1),n(k)}$.

Table 9 compares the imitation accuracy and winrates improvements when switching from the DipNet neural architecture to the one we use (indicating a slight improvement in performance). For our RL experiments we chose the architecture with the highest imitation accuracy despite its decline in winrates; this is because the RL improvement loop relies on imitation, so imitation performance is the chief desideratum.

Furthermore, the winrates are affected by a confounding factor that we uncovered while inspecting the imperfect winrate of the final imitation network against a random policy. What we found was that in games where the network didn't beat the random policy, the network was playing many consecutive all-holds turns, and hitting a 50-year game length limit in our environment. This reflects the dataset: human players sometimes abandon their games, which shows up as playing all-holds turns for the rest of the game. We hypothesize that the encoder that observes the preceding moves-phase board, and especially the actions since then, is better able to represent and reproduce this behaviour. This is to its detriment when playing games, but is easily addressed by the improvement operator.

| | Imitation accuracy (%) | | | | Winrates (%) | | | |
| | Teacher forcing | | Whole-turn | | vs Random | | vs DipNet SL | |
| Architecture | Press | No-press | Press | No-press | 1v6 | 6v1 | 1v6 | 6v1 |
|---|---|---|---|---|---|---|---|---|
| DipNet replication | 56.35 | 58.03 | 25.67 | 26.86 | 100 | 16.67 | 15.99 | 14.42 |
| +Encoder changes | 59.08 | 60.32 | 30.79 | 30.28 | 100 | 16.66 | 16.75 | 14.51 |
| +Relational decoder | 60.08 | 61.97 | 30.26 | 30.73 | 100 | 16.67 | 17.71 | 14.33 |
| −Alliance features | 60.68 | 62.46 | 30.96 | 31.36 | 99.16 | 16.66 | 13.30 | 14.25 |

Table 9: Imitation learning improvements resulting from our changes to DipNet [90].

### C.1 Hyperparameters

For the inputs, we use embedding sizes 10, 16, and 16 for the recent orders $x_o$, the power $p$, and the season $s$, in addition to the same 35 board features per area used in DipNet. The value decoder has a single hidden layer of size 256. The relational order decoder uses an embedding of size 256 for each possible action.

The imitation networks were trained on a P100 GPU, using an Adam optimizer with learning rate 0.003 and batch size 256, on a 50-50 mixture of the No-Press and Press datasets used for DipNet. The datasets were shuffled once on-disk, and were sampled during training via a buffer of $10^4$ timesteps into which whole games were loaded. During training we filtered out powers that don't end with at least 7 SCs as policy targets, and filtered out games where no power attains 7 SCs as value targets. 2000 randomly selected games from each dataset were held out as a validation set.

Power and season embeddings were initialized with random uniform entries in $[-1, 1]$. Previous-order embeddings were initialized with random standard normal entries. The decoder's action embeddings were initialized with truncated normal entries with mean 0 and standard deviation $1/\sqrt{18584}$ (18584 is the number of possible actions). The GNN weights were initialized with truncated normal entries with mean 0 and standard deviation $2/\sqrt{(\text{input width}) \cdot (\text{number of areas})}$. The value network was initialized with truncated standard normal entries with mean 0 and standard deviation $1/\sqrt{\text{input width}}$ for both its hidden layer and its output layer.

### C.2 Data cleaning

We use the dataset introduced in [90] for our pre-training on human imitation data. Before doing so, we run the following processing steps, to deal with the fact that the data is from a variety of sources and contains some errors:

- We exclude any games labelled in the dataset as being on non-standard maps, with fewer than the full 7 players, or with non-standard rules.

- We attempt to apply all actions in the game to our environment. If we do so successfully, we include the game in the dataset. We use observations generated by our environment, not the observations from the dataset, because some of the dataset's observations were inconsistent with the recorded actions.
- As final rewards for training our value function, we take the reward from the dataset. For drawn games, we award $1/n$ to each of the $n$ surviving players.
- We deal with the following variations in the rules (because not every game was played under the same ruleset):
    - Some data sources apparently played forced moves automatically. So if there is only one legal move, and no move for the turn in the dataset, we play the legal move.
    - Some variants infer whether a move is a convoy or land movement from the other orders of the turn, rather than making this explicit. To parse these correctly, we retry failed games, changing move orders to convoys if the country ordering the move also ordered a fleet to perform a convoy of the same route. For example, if the orders are A BEL - HOL, F HOL - BEL and F NTH C BEL - HOL, we update the first order to be A BEL - HOL (via convoy). If this rewritten game can be parsed correctly, we use this for our imitation data.
- Any game which can not be parsed with these attempted corrections is excluded from the dataset. One large class of games that cannot be parsed is those played under a ruleset where the coast of supports matters – that is, a move to Spain (sc) can fail because a support is ordered to Spain (nc) instead. Other than these, the errors were varied; the first 20 games manually checked appear to have units ending up in places which are inconsistent with their orders, or similar errors. It is possible that these involved manual changes to the game state external to the adjudicator, or were run with adjudicators which either had bugs or used non-standard rulesets.

This process resulted in number of exclusions and final data-sets sizes reported in Tab. 10.

|  | Training set | | Validation set | |
|---|---|---|---|---|
|  | Press | No-press | Press | No-press |
| Available games | 104456 | 31279 | 2000 | 2000 |
| Excluded: non-standard map | 833 | 10659 | 17 | 705 |
| Excluded: non-standard rules | 863 | 1 | 19 | 0 |
| Excluded: min SCs not met | 2954 | 517 | 48 | 31 |
| Excluded: unable to parse | 3554 | 79 | 78 | 4 |
| Included | 96252 | 20023 | 1838 | 1260 |

Table 10: Number of games available, included and excluded in our data-sets.

Diplomacy has been published with multiple rulebooks, which slightly differ in some ways. Some rulebooks do not completely specify the rules, or introduce paradoxes. The Diplomacy Adjudicator Test Cases describe a variety of rulesets, interpretations and paradox resolution methods [69]. We make the same interpretations as the webDiplomacy website [71]. Our adjudicator uses Kruijswijk's algorithm [70].

# D  BRPI settings

For all the reinforcement learning runs reported, we use the settings as in section C.1 for the learning settings, with the exception of the learning rate for Adam, which is $10^{-4}$. We update the policy iteration target by adding a new checkpoint every 7.5 million steps of experience. We use an experience replay buffer of size 50000; to save on the amount of experience needed, each datapoint is sampled 4 times before it is evicted from the buffer.

For sampled best response, we use 2 base profiles. We sample 16 candidate moves for each player; in the case where we use two sources of candidate (such as $\pi^{\text{SL}}$ and latest checkpoint), we sample 8

from each. The base profiles are the same for each of the 7 powers, and if the policy being responded to is the same as a policy producing candidates, we reuse the base profiles as candidate moves.

We force draws in our games after random lengths of time. The minimum game length is 2 years; after that we force draws each year with probability 0.05. Since our agents do not know how to agree draws when the game is stalemated, this gives the game length a human-like distribution and incentivises agents to survive even when they are unlikely to win. When games are drawn in this way, we award final rewards dependent on supply centres, giving each power a reward of the proportion of all controlled SCs which they control at the end of the game. Otherwise, no rewards are given except for a reward of 1 for winning.

## E A2C with V-Trace off policy correction

Figure 15: Winrate of 1 A2C v. 6 Imitation Learning (SL). Shaded areas are error-bars over 7000 games, uniformly distributed over the power controlled by A2C. The step counter on the horizontal axis shows the number of state, action, rewards triplets used by the central learner. Note that this is different than other RL plots in the text, where policy iteration loops are shown.

In this section we describe our implementation of the batched advantage actor-critic (A2C) with V-Trace off policy correction algorithm we used as our policy gradient baseline [30], and very briefly comment on its performance. As in our BRPI experiment, we let Reinforcement Learning training start from our Imitation Learning baseline. We use the same network architecture as for BRPI.

Our implementation of A2C uses a standard actor-learner architecture where actors periodically receive network parameters from a central learner, and produce experience using self-play. The single central learner, in turn, retrieves the experience generated by our actors, and updates its policy and value network parameters using batched A2C with importance weighting.

Two points to note in our implementation:

1. Diplomacy's action space is complex: our network outputs an order for each of the provinces controlled by player $p$ at each turn. Orders for each province are computed one by one, and our ROD module ensures inter-province consistency. We therefore expand the off policy correction coefficients for acting policy $\mu$ and target policy $\pi$ as follows: $\rho = \frac{\pi(a_t|s_t)}{\mu(a_t|s_t)} = \frac{p_\pi(a_{t_1}|s_t)p_\pi(a_{t_2}|a_{t_1},s_t)...p_\pi(a_{t_k}|a_{t_1},...,a_{k-1},s_t)}{p_\mu(a_{t_1}|s_t)p_\mu(a_{t_2}|a_{t_1},s_t)...p_\mu(a_{t_k}|a_{t_1},...,a_{k-1},s_t)}$, where $p_\pi(a_{t_l}|a_{t_1},...,a_{t_{l-1}})$ is the probability of selecting order $a_{t_l}$ for unit $l$, given the state of the board at time $t$, $s_t$, and all orders for previous units $a_{t_1},...,a_{t_{l-1}}$, under policy $\pi$.

2. We do not train the value target using TD. Instead, just as in the imitation step of BRPI algorithms, we augment trajectories with returns and use supervised learning directly on this target.

3. As in D, we force draws after random lengths of time, and otherwise only have rewards when games are won. This differs to the A2C agent trained in [90], where a dense reward was given for capturing supply centres.

Fig. 15 shows A2C's performance in 1v6 tournaments against the Imitation Learning starting point (SL). We observe that A2C's win-rate in this setting steadily increases for about 3M steps, and then

gradually declines. In comparisons to other algorithms, we report the performance of the A2C agent with the best win-rate against its Imitation Learning starting point.

# F   Calculation of Confidence Intervals

The confidence intervals in figure 3 are for the mean scores of 5 different experiments. The confidence interval is for the variation in the means across the random seeds. This is calculated based on a normal distribution assumption with unknown variance (i.e. using a t-distribution with 4 degrees of freedom) [68].

The confidence intervals quoted for 1v6 winrate tables are the measurement confidence interval, they only reflect randomness in the outcomes of games between our actual agents, and do not represent differences due to randomness during training the agents. To calculate the confidence interval, we first calculate the confidence interval for the winrate for each combination of singleton agent, 6-country agent, and country the singleton agent plays as. For this confidence interval, we use the Wilson confidence interval (with continuity correction) [125]. We then combine the confidence intervals using the first method from Waller et al. [123]. Unlike using a single Wilson confidence interval on all data, this corrects for any imbalance in the distribution over agents or countries played in the data generated, for example due to timeouts or machine failures during evaluation.

# G   Gradient Descent for Finding an $\epsilon$-Nash Equilibrium in the Meta-Game

Given our definition for an $\epsilon$-Nash, we measure distance from a Nash equilibrium with $\mathcal{L}_{\exp}(\boldsymbol{x}) = \sum_i \mathcal{L}_{\exp_i}(\boldsymbol{x})$, known as *exploitability* or `Nash-conv`, and attempt to compute an approximate Nash equilibrium via gradient descent. By redefining $r^i \leftarrow r^i + \tau \texttt{entropy}(x_i)$, we approach a Quantal Response Equilibrium [82] (QRE) instead; QRE models players with bounded rationality ($\tau \in [0, \infty)$ recovers rational (Nash) and non-rational at its extremes). Further, QRE can be viewed as a method that performs entropy-regularizing of the reward, which helps convergence in various settings [92]. For further discussion of QRE and its relation to convergence of FP methods, see Appendix H.

Computing a Nash equilibrium is PPAD-complete in general [89], however, computing the relaxed solution of an $\epsilon$-Nash equilibrium proved to be tractable in this setting when the number of strategies is sufficiently small.

Note that a gradient descent method over $\mathcal{L}_{\exp}(\boldsymbol{x})$ is similar to, but not the same as Exploitability Descent [80]. Whereas exploitability descent defines a per-player exploitability, and each player independently descends their own exploitability, this algorithm performs gradient descent on the exploitability of the strategy profile ($\mathcal{L}_{\exp}(\boldsymbol{x})$ above), across all agents.

# H   Theoretical Properties of Fictitious Play

Appendix A discusses the relation between the FP variants of BRPI described in Section 3.2 and Stochastic Fictitious Play [38] (SFP). We now investigate the convergence properties of SFP, extending analysis done for two-player games to many-player games (3 or more players). Our key theoretical result is that SFP converges to an $\epsilon$-coarse correlated equilibrium ($\epsilon$-CCE) in such settings (Theorem 1). We make use of the same notation introduced in Appendix A.1.

We first introduce some notation we use. A Quantal best response equilibrium (QRE) with parameter $\lambda$ defines a fixed point for policy $\pi$ such that for all $i$

$$\pi^i(a^i) = \texttt{softmax}\left(\frac{r^i_{\pi^{-i}}}{\lambda}\right) = \frac{\exp\left(\frac{r^i_{\pi^{-i}}(a_i)}{\lambda}\right)}{\sum_j \exp\left(\frac{r^i_{\pi^{-i}}(a_j)}{\lambda}\right)}$$

where $\lambda$ is an inverse temperature parameter and $r^i_{\pi^{-i}}$ is a vector containing player $i$'s rewards for each action $a_i$ given the remaining players play $\pi^{-i}$. The softmax can be rewritten as follows:

$$\frac{\exp\left(\frac{r^i_{\pi^{-i}}(a_i)}{\lambda}\right)}{\sum\limits_j \exp\left(\frac{r^i_{\pi^{-i}}(a_j)}{\lambda}\right)} = \arg\max_{p^i}\left[\langle p^i, r^i_{\pi^{-i}}\rangle + \lambda h^i(p^i)\right]$$

where the function $h^i$ is the entropy. The QRE can then be rewritten as:

$$\max_{p^i}\left[\langle p^i, r^i_{\pi^{-i}}\rangle + \lambda h^i(p^i)\right] = \langle \pi^i, r^i_{\pi^{-i}}\rangle + \lambda h^i(\pi^i).$$

We leverage this final formulation of the QRE in our analysis.

Fictitious Play with best responses computed against this entropy regularized payoff is referred to as *Stochastic* Fictitious Play. Note that without the entropy term, a best response is likely a pure strategy whereas with the entropy term, mixed strategies are more common. The probability of sampling an action $a$ according to this best response is

$$P\left(a = \arg\max_{a^i}\frac{r^i_{\pi^{-i}}(a^i)}{\lambda} + \epsilon_i\right)$$

where $\epsilon_i$ follow a Gumbel distribution ($\mu = 0$ and $\beta = 1$ see the original paper on SFP [38]).

Finally, let $h(p^i)$ denote the entropy regularized payoff of a policy $p^i$. We use the following shorthand notation for the Fenchel conjugate of $h$: $h^*(y) = \max_p\left[\langle p, y\rangle + h(p)\right]$. Note that $h^*(y) = \langle p^*, y\rangle + h(p^*)$ where $p^* = \arg\max_p\left[\langle p, y\rangle + h(p)\right]$ and so therefore, $\frac{dh^*(y)}{dy} = p^*$. This property is used in proving the regret minimizing property of Continuous Time SFP below.

We first recap known results of Discrete Time Fictitious Play and Continuous Time Fictitious Play. As a warm-up, we then prove convergence of Continuous Time Fictitious Play to a CCE.

Next, we review results for Discrete Time Stochastic Fictitious Play and Continuous Time Stochastic Fictitious Play. Lastly, we prove convergence of Continuous Time Stochastic Fictitious Play to an $\epsilon$-CCE.[9]

**Discrete Time Fictitious Play:** Discrete Time Fictitious Play [97] is probably the oldest algorithm to learn a Nash equilibrium in a zero-sum two-player game. The convergence rate is $O\left(t^{-\frac{1}{|A_0|+|A_1|-2}}\right)$ [107] and it has been conjectured in [59] that the actual rate of convergence of Discrete Time Fictitious Play is $O(t^{-\frac{1}{2}})$ (which matches [107]'s lower bound in the 2 action case). A strong form of this conjecture has been disproved in [25].

However Discrete Time Fictitious Play (sometimes referred to as Follow the Leader) is not a regret minimizing algorithm in the worst case (see a counter example in [28]). A solution to this problem is to add a regularization term such as entropy in the best response to get the regret minimizing property (*i.e.* a Follow the *Regularized* Leader algorithm).

**Continuous time Fictitious Play:** The integral version of Continuous Time FP (CFP) is:

$$\pi^i_t = \frac{1}{t}\int\limits_{s=0}^{t} b^i_s ds \quad \text{where } \forall i \text{ and } t \geq 1, b^i_t = \arg\max_{p^i}\left\langle p^i, \frac{1}{t}\int_{s=0}^{t} r^i_{b_s^{-i}} ds\right\rangle,$$

with $b^i_t$ being arbitrary for $t < 1$. A straightforward Lyapunov analysis [47] shows that CFP (in two-player zero-sum) results in a descent on the exploitability $\phi(\pi) = \sum_{i=1}^N \max_{p^i}\langle p^i, r^i_{\pi^{-i}}\rangle - r^i_\pi$. In addition, CFP is known to converge to a CCE in **two** player [88] games.

*Our Contribution:* **Many-Player CFP is Regret Minimizing $\implies$ Convergence to CCE**

We now extend convergence of CFP to a CCE in **many**-player games. Let $r^i_s$ be a measurable reward stream and let the CFP process be:

$$\pi^i_t = \frac{1}{t}\int\limits_{s=0}^{t} b^i_s ds \quad \text{where } \forall i, b^i_t = \arg\max_{p^i}\left\langle p^i, \frac{1}{t}\int\limits_{s=0}^{t} r^i_s ds\right\rangle.$$

Section A.1.1 relates regret to coarse correlated equilibria, so we can prove convergence to a CCE by proving the following regret is sub-linear:

$$Reg((b_s^i)_{s\leq t}) = \max_{p^i} \int_{s=0}^{t} \langle p^i, r_s^i \rangle ds - \int_{s=0}^{t} \langle b_s^i, r_s^i \rangle ds.$$

For all $t \geq 1$ we have:

$$\frac{d}{dt}\left[\max_{p^i} \left\langle p^i, \int_{s=0}^{t} r_s^i ds \right\rangle\right] = \left\langle b_t^i, \frac{d}{dt}\int_{s=0}^{t} r_s^i ds \right\rangle = \langle b_t^i, r_t^i \rangle.$$

We conclude by noticing that:

$$\int_{t=1}^{T} \frac{d}{dt}\left[\max_{p^i}\left\langle p^i, \int_{s=0}^{t} r_s^i ds\right\rangle\right] dt = \int_{t=1}^{T} \langle b_t^i, r_t^i \rangle dt = \int_{t=0}^{T} \langle b_t^i, r_t^i \rangle dt - \int_{t=0}^{1} \langle b_t^i, r_t^i \rangle dt$$

$$= \max_{p^i} \int_{t=0}^{T} \langle p^i, r_t^i \rangle dt - \max_{p^i} \int_{t=0}^{1} \langle p^i, r_t^i \rangle dt.$$

This implies that:

$$\max_{p^i} \int_{t=0}^{T} \langle p^i, r_t^i \rangle dt - \int_{t=0}^{T} \langle b_t^i, r_t^i \rangle dt = \max_{p^i} \int_{t=0}^{1} \langle p^i, r_t^i \rangle dt - \int_{0}^{1} \langle b_t^i, r_t^i \rangle dt.$$

Hence we have $Reg((b_s^i)_{s\leq t}) = O(1)$. This implies that the average joint strategy $\frac{1}{T}\int_{0}^{T} b_t dt$ converges to a CCE (where $b_t(a_1, \ldots, a_N) = b_t^1(a_1) \times \cdots \times b_t^N(a_N)$).

**Discrete time Stochastic Fictitious Play:** Discrete Time Fictitious play has been comprehensively studied [40]. This book shows that Discrete time Stochastic Fictitious Play converges to an $\epsilon$-CCE (this is implied by $\epsilon$-Hannan consistency).

**Continuous time Stochastic Fictitious Play:** The integral version of Continuous Time Stochastic FP (CSFP) is:

$$\pi_t^i = \frac{1}{t}\int_0^t b_s^i ds \quad \text{where } \forall i, b_t^i = \arg\max_{p^i}\left\langle p^i, \frac{1}{t}\int_0^t r_{b_s^{-i}}^i ds + \lambda h^i(p^i)\right\rangle.$$

CSFP is known to converge to a QRE and $\phi_\lambda(\pi) = \sum_{i=1}^{N=2}\max_{p^i}\langle p^i, r_{\pi^{-i}}^i + \lambda h^i(p^i)\rangle - [r_\pi^i + \lambda h^i(\pi^i)] = \sum_{i=1}^{N=2}\max_{p^i}\langle p^i, r_{\pi^{-i}}^i + \lambda h^i(p^i)\rangle - \lambda h^i(\pi^i)$ is a Lyapunov function of the CFP dynamical system [51]. Note that the term $\sum_{i=1}^{N=2} r_\pi^i = 0$ because this is a zero-sum game.

*Our Contribution:* **Many-Player CSFP Achieves Bounded Regret $\implies$ Convergence to $\epsilon$-CCE**

We now present a regret minimization property for CSFP in **many**-player games, which we use to show convergence to an $\epsilon$-CCE. As before, let $r_s^i$ be a measurable reward stream and let the CSFP process be:

$$\pi_t^i = \frac{1}{t}\int_0^t b_s^i ds \quad \text{where } \forall i, b_t^i = \arg\max_{p^i}\left\langle p^i, \frac{1}{t}\int_0^t r_s^i ds + \lambda h^i(p^i)\right\rangle. \tag{4}$$

Note, this process averages best responses to the historical sequence of entropy regularized rewards. We seek to show that the regret of this process with respect to the *un*-regularized rewards grows, at

worst, linearly in $T$ with coefficient dependent on the regularization coefficient, $\lambda$. The result should recover the standard CFP result when $\lambda = 0$.

The proof proceeds by decomposing the regret with respect to the *un*-regularized rewards into two terms: regret with respect to the regularized rewards ($O(1)$) and an $O(T)$ term dependent on $\lambda$ as desired.

Bounding the first term, the regret term, is accomplished by first deriving the derivative of the maximum entropy regularized payoff given historical play up to time $t$ and then recovering the regret over the entire time horizon $T$ by fundamental theorem of calculus.

**Theorem 1.** *CSFP converges to an $\epsilon$-CCE.*

*Proof.* The regret $Reg((b^i)_{s \leq t}) = \max_{p^i} \int_{s=0}^{T} \left[ \langle p^i, r_s^i \rangle \right] ds - \int_{s=0}^{T} [\langle b_s^i, r_s^i \rangle] ds$ is

$$\leq \max_{p^i} \int_{s=0}^{T} \left[ \langle p^i, r_s^i \rangle + \lambda h^i(p^i) - \lambda h^i(p^i) \right] ds - \int_{s=0}^{T} [\langle b_s^i, r_s^i \rangle] ds$$

$$\leq \max_{p^i} \int_{s=0}^{T} \left[ \langle p^i, r_s^i \rangle + \lambda h^i(p^i) \right] ds - T\lambda \min_{p^i} h^i(p^i) - \int_{s=0}^{T} [\langle b_s^i, r_s^i \rangle] ds$$

$$\leq \max_{p^i} \int_{s=0}^{T} \left[ \langle p^i, r_s^i \rangle + \lambda h^i(p^i) \right] ds - T\lambda \min_{p^i} h^i(p^i) - \int_{s=0}^{T} [\langle b_s^i, r_s^i \rangle] ds + \int_{s=0}^{T} [\lambda h^i(b_s^i) - \lambda h^i(b_s^i)] ds$$

$$\leq \max_{p^i} \int_{s=0}^{T} \left[ \langle p^i, r_s^i \rangle + \lambda h^i(p^i) \right] ds - \int_{s=0}^{T} [\langle b_s^i, r_s^i \rangle + \lambda h^i(b_s^i)] ds + \int_{s=0}^{T} \lambda h^i(b_s^i) ds - T\lambda \min_{p^i} h^i(p^i)$$

$$\leq \max_{p^i} \int_{s=0}^{T} \left[ \langle p^i, r_s^i \rangle + \lambda h^i(p^i) \right] ds - \int_{s=0}^{T} [\langle b_s^i, r_s^i \rangle + \lambda h^i(b_s^i)] ds + T\lambda [\max_{p^i} h^i(p^i) - \min_{p^i} h^i(p^i)]$$

$$\overset{L2}{\leq} O(1) + T\lambda [\max_{p^i} h^i(p^i) - \min_{p^i} h^i(p^i)].$$

Section A.1.1 relates regret to coarse correlated equilibria. We thus conclude that CSFP converges to an $\epsilon$-CCE with $\epsilon \leq \lambda[\max_{p^i} h^i(p^i) - \min_{p^i} h^i(p^i)]$.

$\square$

**Lemma 1.** *The derivative of the max entropy regularized payoff up to time $t$ is given by the entropy regularized payoff of the best response:* $\frac{d}{dt} \max_{p^i} \int_{s=0}^{t} \left[ \langle p^i, r_s^i \rangle + \lambda h^i(p^i) \right] ds = \langle b_t^i, r_t^i \rangle + \lambda h^i(b_t^i).$

*Proof.* The derivative is derived by first computing the derivative of the maximum payoff considering the average reward (effectively higher entropy regularization):

$$
\frac{d}{dt}\max_{p^i}\left[\left\langle p^i,\left[\frac{1}{t}\int_{s=0}^{t}r_s^i ds\right]\right\rangle+\lambda h^i(p^i)\right]=\lambda\frac{d}{dt}\max_{p^i}\left[\left\langle p^i,\left[\frac{1}{\lambda t}\int_{s=0}^{t}r_s^i ds\right]\right\rangle+h^i(p^i)\right] \tag{5}
$$

$$
=\lambda\frac{d}{dt}h^{*i}(\frac{1}{\lambda t}\int_{s=0}^{t}r_s^i ds)\overset{\frac{dh^{*i}(y)}{dy}\frac{dy}{dt}}{=}\lambda\langle b_t^i,\frac{d}{dt}\left[\frac{1}{\lambda t}\int_{s=0}^{t}r_s^i ds\right]\rangle=\langle b_t^i,\frac{d}{dt}\left[\frac{1}{t}\int_{s=0}^{t}r_s^i ds\right]\rangle
$$

$$
=\langle b_t^i,\left[-\frac{1}{t^2}\int_{s=0}^{t}r_s^i ds+\frac{1}{t}r_t^i\right]\rangle=\frac{1}{t}\left[\langle b_t^i,r_t^i\rangle-\langle b_t^i,\left[\frac{1}{t}\int_{s=0}^{t}r_s^i ds\right]\rangle\right]
$$

$$
=\frac{1}{t}\left[\langle b_t^i,r_t^i\rangle+\lambda h^i(b_t^i)-\langle b_t^i,\left[\frac{1}{t}\int_{s=0}^{t}r_s^i ds\right]\rangle-\lambda h^i(b_t^i)\right]
$$

$$
=\frac{1}{t}\left[\langle b_t^i,r_t^i\rangle+\lambda h^i(b_t^i)-\max_{p^i}\left[\langle p^i,\left[\frac{1}{t}\int_{s=0}^{t}r_s^i ds\right]\rangle+\lambda h^i(p^i)\right]\right] \tag{6}
$$

where we highlight the use of a special property of the Fenchel conjugate in the second line.

Rearranging equation 5 and equation 6 and then multiplying both sides by $t$ we find:

$$
t\frac{d}{dt}\max_{p^i}\left[\left\langle p^i,\left[\frac{1}{t}\int_{s=0}^{t}r_s^i ds\right]\right\rangle+\lambda h^i(p^i)\right]+\max_{p^i}\left[\left\langle p^i,\left[\frac{1}{t}\int_{s=0}^{t}r_s^i ds\right]\right\rangle+\lambda h^i(p^i)\right]=\langle b_t^i,r_t^i\rangle+\lambda h^i(b_t^i).
$$

As a result we obtain:

$$
\frac{d}{dt}\max_{p^i}\int_{s=0}^{t}\left[\langle p^i,r_s^i\rangle+\lambda h^i(p^i)\right]ds
$$

$$
=\frac{d}{dt}\frac{t}{t}\max_{p^i}\int_{s=0}^{t}\left[\langle p^i,r_s^i\rangle+\lambda h^i(p^i)\right]ds
$$

$$
=\frac{d}{dt}\frac{t}{t}\max_{p^i}\left[\left\langle p^i,\int_{s=0}^{t}r_s^i ds\right\rangle+\int_{s=0}^{t}\lambda h^i(p^i)ds\right]
$$

$$
=\frac{d}{dt}t\max_{p^i}\left[\left\langle p^i,\frac{1}{t}\int_{s=0}^{t}r_s^i ds\right\rangle+\frac{1}{t}\int_{s=0}^{t}\lambda h^i(p^i)ds\right]
$$

$$
=\frac{d}{dt}t\max_{p^i}\left[\left\langle p^i,\frac{1}{t}\int_{s=0}^{t}r_s^i ds\right\rangle+\lambda h^i(p^i)\right]
$$

$$
=t\frac{d}{dt}\max_{p^i}\left[\left\langle p^i,\left[\frac{1}{t}\int_{s=0}^{t}r_s^i ds\right]\right\rangle+\lambda h^i(p^i)\right]+\left[\frac{d}{dt}t\right]\max_{p^i}\left[\left\langle p^i,\left[\frac{1}{t}\int_{s=0}^{t}r_s^i ds\right]\right\rangle+\lambda h^i(p^i)\right]
$$

$$
=\langle b_t^i,r_t^i\rangle+\lambda h^i(b_t^i).
$$

$\square$

**Lemma 2.** *The regret of the stochastic best response process with respect to the entropy regularized payoffs,*

$$
\max_{p^i}\int_{s=0}^{T}\left[\langle p^i,r_s^i\rangle+\lambda h^i(p^i)\right]ds-\int_{s=0}^{T}[\langle b_s^i,r_s^i\rangle+\lambda h^i(b_s^i)]ds,\quad is\ O(1).
$$

*Proof.* Integrating Lemma 1 from 1 to $T$ and decomposing we find

$$\int_1^T \frac{d}{dt}[\max_{p^i} \int_{s=0}^t \left[\langle p^i, r_s^i\rangle + \lambda h^i(p^i)\right] ds]dt = \int_1^T [\langle b_t^i, r_t^i\rangle + \lambda h^i(b_t^i)]dt$$

$$= \int_{s=0}^T [\langle b_s^i, r_s^i\rangle + \lambda h^i(b_s^i)]ds - \int_{s=0}^1 [\langle b_s^i, r_s^i\rangle + \lambda h^i(b_s^i)]ds.$$

Also, note that, by fundamental theorem of calculus, the integral can also be expressed as

$$\int_1^T \frac{d}{dt}[\max_{p^i} \int_{s=0}^t \left[\langle p^i, r_s^i\rangle + \lambda h^i(p^i)\right] ds]dt$$

$$= \max_{p^i} \int_{s=0}^T \left[\langle p^i, r_s^i\rangle + \lambda h^i(p^i)\right] ds - \max_{p^i} \int_{s=0}^1 \left[\langle p^i, r_s^i\rangle + \lambda h^i(p^i)\right] ds.$$

Rearranging the terms relates the regret to a definite integral from 0 to 1

$$\max_{p^i} \int_{s=0}^T \left[\langle p^i, r_s^i\rangle + \lambda h^i(p^i)\right] ds - \int_{s=0}^T [\langle b_s^i, r_s^i\rangle + \lambda h^i(b_s^i)]ds$$

$$= \max_{p^i} \int_{s=0}^1 \left[\langle p^i, r_s^i\rangle + \lambda h^i(p^i)\right] ds - \int_{s=0}^1 [\langle b_s^i, r_s^i\rangle + \lambda h^i(b_s^i)]ds$$

which is $O(1)$ with respect to the time horizon $T$. $\qquad\square$

# I   Size Estimates for Diplomacy

One of the issues that make Diplomacy a difficult AI challenge is the sheer size of the game. We estimate the size of the game of Diplomacy based on No-Press games in the human dataset [90]. This dataset consists of 21,831 games. We play through each game, and inspect how many legal actions were available for each player at each turn of the game.

Some of the games in the dataset are unusually short, with a draw agreed after only a couple of turns. This is usually done to cancel a game on websites without the functionality to do so. We do not attempt to filter such games from our data; as a result, the size estimates here are biased downward.

## I.1   Legal Joint Actions per Turn

Diplomacy turns are in one of three phases: the *movement* (or *Diplomacy*) phase, the *retreats* phase and the *adjustments* phase. The majority of gameplay is in the movement phase, while the retreats and adjustments phases mostly handle the effects of the movement phase. So we consider the size of this movement phase.

In the first turn of diplomacy, there are 22 units on the board, 3 for most players and 4 for Russia. Each unit has approximately 10 legal actions, which can be selected independently of one another. The total number of possibilities for this first turn is $10^{22.3}$.

As the game progresses, the number of units on the board increases to a maximum of 34. Additionally, when units are closer together, there are more opportunities to play *support* moves and as a result the number of legal actions per unit grows. The largest movements phase in our data had a total of $10^{64.3}$ legal action combinations across the 7 players. The median number of possibilities in movement phases is $10^{45.8}$.

Many of these different combinations of actions lead to the same states, for example the action MAR **support** PAR $\rightarrow$ BUR has no affect on the adjudication if the owner of the unit is Paris doesn't take the action PAR $\rightarrow$ BUR, or if the movement to BUR is unopposed, or if another unit moves to MAR, cutting the support.

### I.2   Estimate of the Game Tree Size

We estimate the size of the game tree from a single game by considering the product of the number of legal actions available at each turn of the game. For example, for a game where there were 4 options on the first turn, 2 options on the second, 3 on the third, we estimate the size as $4 \times 2 \times 3 = 12$. The median size was $10^{896.8}$.

Note that as Diplomacy has no repetition rule or turn limit, the game tree is technically infinite. The purpose of our estimate is to give a rough sense of the number of possibilities the agents must consider in any given game. We report the median as the arithmetic mean is dominated by the largest value, $10^{7478}$, which comes from an exceptionally long game (that game lasted 157 movement phases, whereas the median game length was 20 movement phases). When long games are truncated to include only their first 20 movement phases, the median size is $10^{867.8}$, and the maximum is $10^{1006.7}$.

## Footnotes

[6]For further reading on this topic, see [22, 32, 42, 43].

[7]We exclude other phases from this analysis.

[8]An *area* is any location that can contain a unit. This is the 75 provinces (e.g. Portugal, Spain), plus the 6 coastal areas of the three bicoastal provinces (e.g. Spain (south coast), Spain (north coast)).

[9]This result generalizes the result in the warm-up.