[Reviews · NeurIPS 2020]

Review 1

Summary and Contributions: This paper proposes a sampled-best response policy iteration algorithm to train the agents in a multi-player game (Diplomacy: 7-player board game) with simultaneous moves. Apart from proposing the algorithm for challenging multi-player games, simultaneous moves with a large combinatorial action space, the authors also provide game theoretic equilibrium analysis, exploitability of agents as well as provide extensive ablation experiments on smaller problems that they can compute exact best response, Blotto. I have read the response and other reviewers' comments. My opinion has not changed.

Strengths: This paper successfully adopts Reinforcement Learning (Policy Iteration) to Diplomacy. The authors gave a good motivation why this problem is important and how it is a huge step-up from the RL for commonly explored 2-player games. They propose an approximate iterated best response method which is also tailored for simultaneous moves by considering mixed strategies. Moreover, they improved the existing supervised learning work for Diplomacy (DipNet) so that it can be used with RL procedures. The authors proposed three variants Fictitious Play Policy Iteration (FPPI-1, FPPI-2) and Iterated Best Response discussing the drawbacks of the initial proposal FPPI-1. The authors also include extensive theoretical analysis on equilibrium (Nash, Corse Correlated), exploitability and ablation study for the proposed algorithms with a smaller but similar problem in the Appendix. For experiments, the authors also performed extensive evaluation by using 4 metrics. Lastly, the future impact for the work is promising especially for the problems with many agents that needs collaboration even in a competitive setting as well as simultaneously moving games prominently StarCraft, Dota, controlling of cars, robots for more complex tasks.

Weaknesses: In section 2.1, when the paper describes the game Diplomacy, there are some fats like each unit has 10 legal actions are left out. I came to know this when I read the Appendix, but it would be better if it is in main paper so that we can imagine how large the problem is. The paper has presented effects on CCEDist with the different B and C values for Colonel Blotto in Appendix A.4.2 but the authors did not do ablation studies on approximated best response on the problem they proposed which is Diplomacy. It would be better if the authors can present the effects of different B/C values for Diplomacy. There is a typo in line 316: it should be “suggesting our agents to use fully mixed strategies.” I also find the proposed approach is somehow straightforward.

Correctness: The claims and methods used in the paper is correct according to the empirical methodology described in the paper and Appendix. Some additional questions: 1. For the diagonal entries of Table 1, you have noted in footnote that the average scores are not necessarily 1/7. However, can you provide some intuition on why only IBR and FPPI-2 have less than 1/7=14.28%? Because it seems like even when we are using agents with same algorithm, we cannot expect that our agent is the one will not lose (since there will be an agent which gets better score than average which is 14.28%)? 2. In Appendix A.2.4 where you have investigated B and C values for Blotto, the distance to equilibria only became better (smaller) when B=64 and C larger than at least 16. This is for smaller problem of Blotto so intuitively B, C should be around this value for Diplomacy. However, in Appendix D, you set B=2, C=16 for Diplomacy. Can you elaborate more on the best response algorithm with these values? What is the guarantee that the best response algorithm with given B, C values will give the best response? Is there any lower bound for the reward from Q of best response algorithm to the real maximum reward possible for each agent of each step?

Clarity: The paper is well-written, and the details are well-explained in the Appendix.

Relation to Prior Work: The authors clearly discussed that this work is a step-up from regularly explored problem of 2-player reinforcement learning problems by solving multi-player games having both cooperative and competitive agent decisions with simultaneous moves. The authors also described clearly that the state-of-the-art supervised-learning solution of Diplomacy, DipNet does not work well with A2C (actor-critic) to be readily adopted to be used in RL approximate best response setting. Hence, the authors gave a clear motivation on the problem proposed, the solution used, and the extensions proposed from SOTA in the paper.

Reproducibility: Yes

Additional Feedback:


Review 2

Summary and Contributions: Motivated by the diverging reviewing scores I set out to carefully re-evaluate the article. My initial verdict was tainted by the authors strong emphasis on the game Diplomacy, and the lack of providing intuition for the contribution embedding. In essence, the proposed method improves a joint policy with Monte Carlo estimates over follow-up states, whose value is also learned by the same network that produces this joint policy. Monte Carlo sampling and 1-turn improvements are proposed for computational tractability. The algorithm does provide plenty of references for related work on general AI, Diplomacy and BRPI approaches. Empirical evaluation is sound and does provide insights into transitivity and exploitability. It is apparent that constructing a working learning system for Diplomacy was a large effort, and state of the art techniques have been improved to do so. Yet overall, the transfer insights are not spelled out clearly for the reader. Particular responses to the rebuttal: - While transferable insights are somewhat apparent in the abstract, emphasis of title, introduction, background, and conclusions is on Diplomacy rather than the interesting contribution, which got me side-tracked. - I would have loved to see the arguments of (2) R2 inside the original submission; they resolve my concerns in that direction. Open questions: - Algorithm 2 lines 5 and 6: should they read subscript i or t? ------ This article presents a tailor-made algorithm for No-Press Diplomacy, a 7-player game that is designed to yield competition-cooperation tradeoffs. Empirical evaluation present improvements over state of the art in two performance metrics (in head-to-head and against a population), and two policy progression metrics (transitivity and exploitability).

Strengths: The learning system achieves state-of-the-art performance in the game, bringing together several techniques of sampling and training neural networks through policy improvement.

Weaknesses: The contribution is very domain-specific, limiting the target audience at NeurIPS. While some aspects of the method may transfer, they are not juxtaposed sufficiently to previous work to assess or appreciate their novelty. Overall, the level of reflection could be deeper. For example, the game tree complexity is mentioned as a challenge to find good solutions, but smoothness of performance over this large space is not identified as the underlying hardness factor for finding good policies (especially by such a sampling-heavy approach as it is applied here). At another point, the authors mention that the ambition is to "reduce variance" of estimates, but this is not further elevated to a more general principle that guides and motivates methodological choices.

Correctness: There are some imprecisions and omissions of background: The Q-equation on page 4 (line 140) supposes a game model that permits only an outcome and no intermediate rewards. In the absence of any task formalism in the Background section, this comes as a surprise. The algorithms make inconsistent use of notation; in particular sample numbers receive seemingly arbitrary letters, and actions are denoted b_j where I would expect a_{-i} and c_j where I would expect a_i. In addition, the Q-estimator is now only parameterised by the action, rather than the state-action pair. Further, the Sampled Best Response returns an action, rather than a policy. Since a Best Response refers to a policy, this is conceptually confusing.

Clarity: The presentation is in good language, and overall reasonable to follow. Yet, the Background does not cover expected material (task setting, e.g. Markov games), and section 2.1 stands as a lone section. Section 3.1 first reads as if Sampled Best Response computes an improvement over ALL states (like dynamic programming); this seems to later be rectified by describing its use in policy iteration in Section 3.2, which generates trajectories (over which I assume this improvement is calculated). "Therefore we use Monte-Carlo sampling." Should be made more precise: of what, maybe reference an equation, or re-state parts of it.

Relation to Prior Work: Section 3.1-3.2 present the potentially most transferable part of the method without crisp demarcation to previous or alternate approaches.

Reproducibility: Yes

Additional Feedback: "NFSP" is an abbreviation that seems to not be introduced. - "The figure on the right is identical" add "in setup"; as the figures are clearly not identical. - Why use transitivity and not ELO ranking?


Review 3

Summary and Contributions: This paper introduces a few policy iteration approaches to improve agent performance in no-press Diplomacy. The agent trains by repeatedly calculating best responses to previous versions (or the average of previous versions) of itself. They show that when their algorithm is initialized from the prior state of the art agent, it learns to beat that agent. They also evaluate the agent with a number of other metrics.

Strengths: The authors present a thorough analysis of their techniques and compare to prior agents. The techniques themselves are not extremely novel, but most research in this field has focused on either two-player zero-sum games or simple multi-agent games, so it is valuable to see these techniques evaluated in a complex multi-agent domain like Diplomacy. The paper is very well written and easy to follow.

Weaknesses: I'm concerned that the comparison to DipNet, the prior state of the art, is misleading because the authors initialize their algorithm by effectively computing a best response to DipNet. (Specifically, they start by computing a best response to a supervised learning agent that is trained on the same dataset as DipNet.) Since they beat DipNet, the authors say that they are "stronger" than DipNet. However, beating DipNet is expected if one were to compute a best response to DipNet, even if the best response is a "weaker" policy. To illustrate why this is a problem, one could imagine a situation like Rock-Paper-Scissors where DipNet is biased toward playing Rock, so the techniques introduced in this paper effectively learn to always choose Paper. Paper beats Rock, but one is not "stronger" than the other. Of course this is not a perfect analogy because the authors then proceed to compute additional best responses, but it is possible that these iterative best responses find a local minimum that does particularly well against DipNet because that is where it is initialized. There are some ways to get a better indication of whether these new agents are indeed "stronger" than DipNet. One would be to look at exploitability (which is certainly not a perfect metric, but it is one metric). Another would be to look at head-to-head performance with other independently developed agents (again not a perfect metric, but certainly a useful one). For exploitability, it appears that the supervised learning agent may have *lower* exploitability than the RL agents (based on appendix B.2). Am I misunderstanding these results? There appears to be some room for interpretation in these results. As for comparison to a separate agent, none is presented in the paper. The original DipNet paper compared to Albert, a rule-based agent that was the previous state of the art. It would be very valuable to also include comparisons to Albert in this paper to strengthen the case that these techniques are actually improving the supervised learning policy. If this new agent were to beat Albert by more than DipNet beat Albert, that would be fairly convincing evidence that the new agent is indeed stronger than DipNet. For this reason, I would appreciate it if the authors could evaluate against Albert and report those results in the author feedback if at all possible. At the very least, the authors should explain this issue in the paper, and it would be more appropriate to say that the agents in this paper "beat" DipNet, rather than say they are "stronger" than DipNet.

Correctness: The methods are correct from what I can tell. The claims are mostly correct, though there is the concern I raised above.

Clarity: The paper is very well written.

Relation to Prior Work: The authors do a good job covering related work. That said, the number of citations, at over 100, may be a bit excessive and some parts of the paper (especially the first paragraph of section 2) feel like the authors are just listing every paper that might be even slightly related.

Reproducibility: Yes

Additional Feedback: In the first sentence, the authors list a bunch of major AI results in games. For chess, the authors point to the AlphaZero paper. It would be more appropriate to point to Campbell et al. here (which the authors later do in Section 2). On line 37 the authors say "In this paper we focus on learning strategic interactions in a many-agent setting, so we consider the popular No Press variant, where no explicit communication is allowed." This seems to imply that the authors focus on the No Press variant *because* the paper focuses on learning strategic interactions in a many-agent setting. But I don't see why the No Press variant would be better suited for that. On line 79 the authors add a caveat that the many-player poker result used expert abstraction and end-game solving. I don't see why those are issues. In my opinion, the point about players folding early and prohibition on collusion is the important point to make. The insertion of a comma after "game" and before "until" makes that sentence difficult to parse though and seems to imply that collusion is strictly prohibited specifically when only two players remain. The authors say they compute a best response to a random iteration. Why not try weighted averages? In the head-to-head comparison, IBR and FPPI-2 achieve only 12.9% and 12.7% against themselves. This seems too far below 1/7 to be due to variance. Is there something else going on here? What is the score for ties? The right side of Figure 3 was a bit difficult to interpret. ===POST-FEEDBACK COMMENTS=== Thank you for running the Albert experiments in such a short time window. I agree the results convincingly show that this agent is stronger than DipNet and I am now more confident that the paper should be accepted. I still think it would be worth discussing in the paper the best response issue related to the DipNet head-to-head experiments. I think it would also be interesting to see how the performance against DipNet/Albert evolves over time. One might expect performance against DipNet to peak early as the policy overfits to a best response against DipNet and then decrease over time as it shifts away from being a best response to DipNet. Regarding exploitability calculation, it seems to me like a better approach would be to train an RL agent against six fixed FPPI-2 or SL agents. I think that would find a better response than the approach you are using and the results would be more directly comparable between different agents. I don't think I understand the authors' explanation for why some diagonal entries in the matrix are below 1/7th. If the authors are playing one FPPI-2 run against a different FPPI-2 run, then that should be fixed so that they are using the same run for all entries in the matrix (or, even better, averaging over all runs). The authors should also explain what their scoring system is in case of ties.


Review 4

Summary and Contributions: Thank you for the rebuttal. While not all my points were addressed in the author response, I was already quite happy with this paper, and after rebuttal and discussion I see no reason to lower my score. This paper proposes a new approach to No-Press (no communication) Diplomacy, a 7 player non-cooperative game which models sequential social dilemmas. The approach uses a sampling based approximate best response method to improve over DipNet, the earlier state-of-the-art in Diplomacy.

Strengths: - This paper has a very impressive literature review. - The work tackles a game that is not a two-player zero-sum game - The proposed approaches (BRI, FPPI-1, FPPI-2) all have higher average scores on Diplomacy than DipNet, the earlier state-of-the-art approach. - There is an extensive empirical evaluation from multiple angles: head to head for comparing algorithms, an agent versus a population of other agents, choosing a ‘champion’ agent, and exploitability.

Weaknesses: The paper was specifically written for one game (Diplomacy), and specifically one variation (no-press) of that game. While I think this work is of interest to the NeurIPS community, the current scope is inherently limited.

Correctness: I have not found any major flaws.

Clarity: Overall the paper is very well written. Some questions and suggestions: - It would be helpful to include a reference or explanation on Best-response calculations. - Line 180: What do you mean with “historical network checkpoints”? Are those previous versions of the network? - Section Policy Transitivity: Can you explain why it is zero-sum? - Line 165: Can you explain or give an example of why this happens with exact best responses? - I would like the authors to add some details to their section on exploitability, instead of having only a teaser in the paper and the details in the Appendix.

Relation to Prior Work: - The literature study is very extensive, both in terms of research on AI + games and diplomacy + AI in general. - There is some relevant work on learning mixtures of GANs, specifically the deep learning variant (Oliehoek 2018) of the (Parallel) Nash Memory (Ficici & Pollack, 2003). I would like the authors to explain what the difference is of using the SBR approach (especially with the sampling of historical checkpoints in FPPI-1) over the Nash Memory approach, which also preserves older strategies. References: Ficici, Sevan G., and Jordan B. Pollack. "A game-theoretic memory mechanism for coevolution." 2003. Oliehoek, Frans A., et al. "Beyond local nash equilibria for adversarial networks." 2018.

Reproducibility: Yes

Additional Feedback: - The authors have clearly put much thought and effort into their broader impact section, which is appreciated.

[Author Response · NeurIPS 2020]

Thanks for the thoughtful comments and feedback. We are pleased that most reviewers agreed that work on >2player games is needed: **R1** wrote 'this problem is important and ... a huge step-up from ... 2 player games', and [**R3**, **R4**] say the >2 player domain is a strength of the paper. All reviews commented that we used multiple evaluation metrics, with positive feedback on the thoroughness of our analysis. We feel this is crucial in mixed-motive settings, where no single metric fully captures performance, so we made a special effort here - thanks for recognising this.

**R3** raises an important point about whether starting with SL means results against DipNet might be misleading. For the author feedback **we ran the suggested experiment using Albert** to resolve this. Using Albert is tricky: it is slow and only available as a Windows binary using DAIDE. We replicated, within statistical error, DipNet's result: **1 DipNet won** $0.39 \pm 0.06$ **vs 6 Alberts** (winrate$\pm 95\%$ CI). We only had time to test one of our methods; **FPPI-2 won much more vs 6 Alberts** $0.75 \pm 0.05$. For the camera-ready paper, we'll add results for all algorithms vs Albert to Table 1.

**R2** wrote: 'The contribution is very domain-specific, limiting the target audience at NeurIPS'. **We disagree:** (1) As **R4** said, research specific to Diplomacy is still of interest to the NeurIPS community. (2) SBR, FPPI, and our evaluation methods are novel and general. (3) We present novel empirical results relevant to other domains and algorithms.

(1) Diplomacy is an **established challenge** for the AI community, with many papers in AI conferences since the 1980s, including the (very domain specific) DipNet paper at NeurIPS last year. Most of our reviewers felt this domain was of interest, so we are confident that many in the NeurIPS audience will be interested in the work.

(2) **R2** wrote 'Section 3.1-3.2 present the potentially most transferable part of the method without crisp demarcation...'. We used ideas in previous works, with new ideas to scale to Diplomacy. *NFSP and PSRO* approximate Fictitious Play (FP) using RL, applying model-free RL to produce best responses (BRs). But *DQN* (used by NFSP) requires a small action space, and *A2C* is ineffective in Diplomacy. In contrast, we use SBR to produce best responses without running an RL algorithm in the inner loop. This made FPPI-1/2 possible, the first Policy Iteration (PI) methods to approximate FP. *ExIt and AlphaZero* also use PI, but their MCTS requires sequential moves and only $\sim 100$s of actions per turn. *CFR* (e.g. [18,81]) handles simultaneous moves, but require enumeration of the joint action space. SBR is more general as it handles simultaneous moves and larger action spaces. We'll add a detailed discussion to highlight these points.

In summary, previous self-play RL methods cannot cope with the action space or simultaneous moves, and have rarely been studied in the general many-agent case. These features characterise **many domains** such as large scale fleet management, multi-commodity markets and multi-robot control (among the further domains that **R1** pointed out). Finally, our evaluation methods, e.g. Nash based policy transitivity, are almost entirely general to many-agent settings.

(3) Challenge domains like Go, Poker, or Diplomacy are useful since they tell us what kinds of methods work. Insights from domain specific methods, e.g. AlphaGo lead to more general methods, e.g. AlphaZero. **Key takeaways** from this work are: sampling-heavy best response (BR) estimation is sufficient to tackle large-scale many-player environments; stochastic BRs improve the convergence of IBR; and that FPPI-2's method of averaging policies is more effective than FPPI-1's NFSP-style method. **We'll emphasize these contributions**, of interest to the wider MARL community.

**R1** suggests we study the effects of B/C values in Diplomacy. We agree this is interesting and will add ablations on it.

**R1** and **R3** asked about diagonal entries of Table 1. In 1v6, the '6' are identical, i.e. from the same training run. So a '1' player from a different training run has a slight disadvantage, being further from their training distribution. For BRPI methods, we used multiple training runs, this is why for IBR and FPPI-2 the '1' player had a winrate below 1/7th.

**R3** asked about **the exploitability of the SL agent**. For RL agents we use their own value function to get a very specific exploit. SL's value net is very inaccurate, so we used a worse exploiter vs SL, using an *arbitrary* BRPI value function. Comparing these exploits between SL and BRPI confounds exploiter quality and exploitability. We have since tried different BRPI value nets to exploit SL, showing we can exploit SL by more than the RL agents. Few-shot exploitability does not have this issue, enabling better between-algorithm comparisons. We'll add a discussion of this.

Answers to specific questions: **R1**: SBR, while effective, has no theoretical guarantee: there is a game with $\leq 2(B+2)$ actions where SBR with $B$ base profiles prefers a suboptimal action vs the Nash. We can add this result with proof to an appendix. **R2**: We discussed in section 2.1 how different unit moves are interdependent. This implies that performance is **not smooth** over the action space. Elo *assumes* transitivity to model skill, in contrast we are *testing* whether transitivity in fact holds. **R3**: We agree with the comment on many-player Poker. Weighted averages in BRPI are interesting future work. **R4**: Yes, 'Historical network checkpoints' are previous versions (parameters) of the network. Meta-games are 0-sum as players' rewards are equal to the rewards in Diplomacy, which is (ultimately) 0-sum. Appendix A.3 gives an example of bad behaviour from exact BRs in IBR, we will refer to it on line 165. We will add more detail on exploitability to the main text.

We thank the reviewers for the positive feedback on clarity, discussion of related work and broader impact section, and are grateful for the suggestions on ways to improve these further, which we will incorporate in the final version.

[Meta-Review · NeurIPS 2020]

The paper proposes an approach for complex 7-player multi-agent game Diplomacy (no-communication). All the reviewers liked the clear writing and thorough evaluation. Some concerns were raised about fairer comparison to prior state-of-the-art which were partially addressed by the authors' rebuttal. The paper was discussed among the reviewers and everyone agrees that the paper has valuable insights to be shared with community. Please incorporate appropriate changes in the camera ready version of paper in response to reviewers' final comments. For instance, clarifying the comparison setup against prior state-of-the-art (DipNet) that the approach is initialized by effectively computing a best response to baseline, etc. Please refer to reviews for more details.